# Diffusion Policies Creating a Trust Region for Offline Reinforcement Learning

**Tianyu Chen**    **Zhendong Wang**    **Mingyuan Zhou**
The University of Texas at Austin
{tianyuchen, zhendong.wang}@utexas.edu
mingyuan.zhou@mccombs.utexas.edu

## Abstract

Offline reinforcement learning (RL) leverages pre-collected datasets to train optimal policies. Diffusion Q-Learning (DQL), introducing diffusion models as a powerful and expressive policy class, significantly boosts the performance of offline RL. However, its reliance on iterative denoising sampling to generate actions slows down both training and inference. While several recent attempts have tried to accelerate diffusion-QL, the improvement in training and/or inference speed often results in degraded performance. In this paper, we introduce a dual policy approach, Diffusion Trusted Q-Learning (DTQL), which comprises a diffusion policy for pure behavior cloning and a practical one-step policy. We bridge the two polices by a newly introduced diffusion trust region loss. The diffusion policy maintains expressiveness, while the trust region loss directs the one-step policy to explore freely and seek modes within the region defined by the diffusion policy. DTQL eliminates the need for iterative denoising sampling during both training and inference, making it remarkably computationally efficient. We evaluate its effectiveness and algorithmic characteristics against popular Kullback–Leibler divergence-based distillation methods in 2D bandit scenarios and gym tasks. We then show that DTQL could not only outperform other methods on the majority of the D4RL benchmark tasks but also demonstrate efficiency in training and inference speeds. The PyTorch implementation is available at `https://github.com/TianyuCodings/Diffusion_Trusted_Q_Learning`.

## 1 Introduction

Reinforcement learning (RL) centers on developing a policy to make sequential decisions by interacting with an environment, aiming to maximize the total rewards accumulated over a trajectory [Wiering and Van Otterlo, 2012, Li, 2017]. Offline RL addresses these challenges by enabling the training of an RL policy from fixed datasets of previously collected data, without further interactions with the environment [Lange et al., 2012, Fu et al., 2020]. This approach leverages large-scale historical data, mitigating the risks and costs associated with live environment exploration. However, offline RL introduces its own set of challenges, primarily related to the distribution shift between the data on which the policy was trained and the data it encounters during evaluation [Fujimoto et al., 2019]. Additionally, the limited expressive power of policies that may not adequately capture the multimodal nature of action behaviors is also a concern.

To mitigate distribution shifts, popular approaches include weighted regression, such as IQL [Kostrikov et al., 2021] and AWAC [Nair et al., 2020], aimed at extracting viable policies from historical data. Alternatively, behavior-regularized policy optimization techniques are employed to constrain the divergence between the learned and in-sample policies during training [Wu et al., 2019]. Notable examples of this strategy include TD3-BC [Fujimoto and Gu, 2021], CQL [Kumar et al.,

38th Conference on Neural Information Processing Systems (NeurIPS 2024).

2020], and BEAR [Kumar et al., 2019]. These methods primarily utilize either Gaussian or deterministic policies, which have faced criticism for their limited expressiveness. Recent advancements have incorporated generative models to enhance policy representation. Variational Autoencoders (VAEs) [Kingma and Welling, 2013] and Generative Adversarial Networks (GANs) [Goodfellow et al., 2020] have been introduced into the offline RL domain, leading to the development of algorithms such as BCQ [Fujimoto et al., 2019] and GAN-Joint [Yang et al., 2022]. Moreover, diffusion models have recently emerged as the most prevalent tools for achieving expressive policy frameworks [Janner et al., 2022, Wang et al., 2022a, Chen et al., 2023, Hansen-Estruch et al., 2023, Chen et al., 2022], demonstrating state-of-the-art performance on the D4RL benchmarks. Diffusion Q-Learning (DQL) [Wang et al., 2022a] applies these policies for behavior regularization, while algorithms such as IDQL [Hansen-Estruch et al., 2023] leverage diffusion-based policies for policy extraction.

However, optimizing diffusion policies for rewards in RL is computationally expensive due to the need for iteratively denoising to generate actions during both training and inference. Recently, distillation has become a popular technique for reducing the computational costs of diffusion models, *e.g.*, score distillation sampling (SDS) [Poole et al., 2022] and variational score distillation (VSD) [Wang et al., 2024] in 3D generation, and Diff-Instruct [Luo et al., 2024], Distribution Matching Distillation [Yin et al., 2023], and Score identity Distillation (SiD) [Zhou et al., 2024] in 2D. These advancements distill the iterative denoising process of diffusion models into a one-step generator. SRPO [Chen et al., 2023] employs SDS [Poole et al., 2022] in the offline RL field by incorporating a Kullback–Leibler (KL) divergence-based behavior regularization loss to reduce training and inference costs. Another related work, IDQL [Hansen-Estruch et al., 2023], selects action candidates from a diffusion behavior-cloning policy and requires a 5-step iterative denoising process to generate multiple candidate actions (ranging from 32 to 128) during inference, which remains computationally demanding. Unlike previous approaches, our paper introduces a diffusion trust region loss that moves away from focusing on distribution matching; instead, it emphasizes establishing a safe, in-sample behavior region. We then simultaneously train dual policies: a diffusion policy for pure behavior cloning and a one-step policy for actual deployment. The one-step policy is optimized based on two objectives: the diffusion trust region loss, which ensures safe policy exploration, and the maximization of the Q-value function, guiding the policy to generate actions in high-reward regions. We elucidate the differences between our diffusion trust region loss and KL-based behavior distillation in Section 3 empirically and theoretically. Our method consistently outperforms KL-based behavior distillation approaches. We provide more discussions on related work in Appendix A.

In summary, we propose DTQL with a diffusion trust region loss. DTQL achieves new state-of-the-art results in majority of D4RL [Fu et al., 2020] benchmark tasks and demonstrates significant improvements in training and inference time efficiency over DQL [Wang et al., 2022a] and related diffusion-based methods.

## 2 Diffusion Trusted Q-Learning

Below, we first introduce the preliminaries of offline RL and basics of diffusion policies for our modeling. We then propose a new diffusion trust region loss which inherently avoids exploring out-of-distribution actions and hence enables safe and free policy exploration. Finally, we introduce our algorithm Diffusion Trusted Q-Learning (DTQL), which is efficient and well-performed.

### 2.1 Preliminaries

In RL, the environment is typically defined within the context of a Markov Decision Process (MDP). An MDP is characterized by the tuple $M = \{S, \mathcal{A}, p_0(\boldsymbol{s}), p(\boldsymbol{s}'|\boldsymbol{s}, \boldsymbol{a}), r(\boldsymbol{s}, \boldsymbol{a}), \gamma\}$, where $S$ denotes the state space, $\mathcal{A}$ represents the action space, $p_0(\boldsymbol{s})$ is the initial state distribution, $p(\boldsymbol{s}'|\boldsymbol{s}, \boldsymbol{a})$ is the transition kernel, $r(\boldsymbol{s}, \boldsymbol{a})$ is the reward function, and $\gamma$ is the discount factor. The objective is to learn a policy $\pi_\theta(\boldsymbol{a}|\boldsymbol{s})$, parameterized by $\theta$, that maximizes the cumulative discounted reward $\mathbb{E}\left[\sum_{t=0}^{\infty} \gamma^t r(\boldsymbol{s}_t, \boldsymbol{a}_t)\right]$. In the offline setting, instead of interacting with the environment, the agent relies solely on a static dataset $\mathcal{D} = \{\boldsymbol{s}, \boldsymbol{a}, r, \boldsymbol{s}'\}$ collected by a behavior policy $\mu_\phi(\boldsymbol{a}|\boldsymbol{s})$. This dataset is the only source of information for the agents.

## 2.2 Diffusion Policy

Diffusion models are powerful generative tools that operate by defining a forward diffusion process to gradually perturb a data distribution into a noise distribution. This model is then employed to reverse the diffusion process, generating data samples from pure noise. While training diffusion models is computationally inexpensive, inference is often costly due to the need for iterative refinement-based sequential denoising. In this paper, we only train a diffusion model and avoid using it for inference, thus significantly reducing both training and inference times.

The forward process involves initially sampling $x_0$ from an unknown data distribution $p(x_0)$, followed by the addition of Gaussian noise to $x_0$, denoted by $x_t$. The transition kernel $q_t(x_t|x_0)$ is given by $x_t = \alpha_t x_0 + \sigma_t \varepsilon$, where $\alpha_t$ and $\sigma_t$ are predefined, and $\varepsilon$ represents random Gaussian noise.

The objective function of the diffusion model aims to train a predictor for denoising noisy samples back to clean samples, represented by the optimization problem:

$$\min_\phi \mathbb{E}_{t,x_0,\varepsilon\sim\mathcal{N}(0,I)}[w(t)\|\mu_\phi(x_t,t) - x_0\|_2^2] \tag{1}$$

where $w(t)$ is a weighted function dependent only on $t$. In offline RL, since our training data is state-action pairs, we train a diffusion policy using a conditional diffusion model as follows:

$$\mathcal{L}(\phi) = \mathbb{E}_{t,\varepsilon\sim\mathcal{N}(0,I),(a_0,s)\sim\mathcal{D}}[w(t)\|\mu_\phi(a_t,t|s) - a_0\|_2^2] \tag{2}$$

where $a_0, s$ are the action and state samples from offline datasets $\mathcal{D}$, and $a_t = \alpha_t a_0 + \sigma_t \varepsilon$. Following previous work [Chen et al., 2023, Hansen-Estruch et al., 2023, Wang et al., 2022a], $\mu(a_t, t|s)$ can be considered an effective behavior-cloning policy.

**The ELBO Objective** The diffusion denoising loss is intrinsically connected with the evidence lower bound (ELBO). It has been demonstrated in prior studies [Ho et al., 2020, Song et al., 2021, Kingma et al., 2021, Kingma and Gao, 2024] that the ELBO for continuous-time diffusion models can be simplified to the following expression (adopted in our setting):

$$\log p(a_0|s) \geq \text{ELBO}(a_0|s) = -\frac{1}{2}\mathbb{E}_{t\sim\mathcal{U}(0,1),\varepsilon\sim\mathcal{N}(0,I)}\left[w(t)\|\mu_\phi(a_t,t|s) - a_0\|_2^2\right] + c, \tag{3}$$

where $a_t = \alpha_t a_0 + \sigma_t \varepsilon$, $w(t) = -\frac{d\text{SNR}(t)}{dt}$, and the signal-to-noise ratio $\text{SNR}(t) = \frac{\alpha_t^2}{\sigma_t^2}$, $c$ is a constant not relevant to $\phi$. Since we always assume that the $\text{SNR}(t)$ is strictly monotonically decreasing in $t$, thus $w(t) > 0$. The validity of the ELBO is maintained regardless of the schedule of $\alpha_t$ and $\sigma_t$.

Kingma and Gao [2024] generalized this theorem stating that if the weighting function $w(t) = -v(t)\frac{d\text{SNR}(t)}{dt}$, where $v(t)$ is monotonic increasing function of $t$, then this weighted diffusion denoising loss is equivalent to the ELBO as defined in Equation 3. The details of how to train the diffusion policy, including the weight and noise schedules, will be discussed in Section 4.3.

## 2.3 Diffusion Trust Region Loss

We found that optimizing diffusion denoising loss from the data perspective with a fixed diffusion model can intrinsically disencourage out-of-distribution sampling and lead to mode seeking. For any given $s$ and a fixed diffusion model $\mu_\phi$, the loss is to find the optimal generation function $\pi_\theta(\cdot|s)$ that can minimize the diffusion-based trust region (TR) loss:

$$\mathcal{L}_{\text{TR}}(\theta) = \mathbb{E}_{t,\varepsilon\sim\mathcal{N}(0,I),s\sim\mathcal{D},a_\theta\sim\pi_\theta(\cdot|s)}[w(t)\|\mu_\phi(\alpha_t a_\theta + \sigma_t\varepsilon, t|s) - a_\theta\|_2^2], \tag{4}$$

where $\pi_\theta(a|s)$ is a one-step generation policy, such as a Gaussian policy.

**Theorem 1.** *If policy $\mu_\phi$ satisfies the ELBO condition of Equation 3, then the Diffusion Trust Region Loss aims to maximize the lower bound of the distribution mode $\max_{a_0}\log p(a_0|s)$ for any given $s$.*

*Proof.* For any given state $s$

$$\max_{a_0}\log p(a_0|s) \geq \max_\theta\mathbb{E}_{a_\theta\sim\pi_\theta(\cdot|s)}\left[\log p(a_\theta|s)\right] \geq \max_\theta\mathbb{E}_{a_\theta\sim\pi_\theta(\cdot|s)}\left[\text{ELBO}(a_\theta|s)\right]$$

$$= \min_\theta\frac{1}{2}\mathbb{E}_{t\sim\mathcal{U}(0,1),\varepsilon\sim\mathcal{N}(0,I),a_\theta\sim\pi_\theta(\cdot|s)}\left[w(t)\|\mu_\phi(\alpha_t a_\theta + \sigma_t\varepsilon, t|s) - a_\theta\|_2^2\right] + c$$

Then, during training, we consider all states $s$ in $\mathcal{D}$. Thus, by taking the expectation over $s \sim \mathcal{D}$ on both sides and setting $t \sim \mathcal{U}(0,1)$, we derive the loss described in Equation 4. $\square$

By definition of the mode of a probability distribution, we know minimizing the loss given by Equation 4 aims to maximize the lower bound of the mode of a probability. Unlike other diffusion models that generate various modalities by optimizing $\phi$ to learn the data distribution, our method specifically aims to generate actions (data) that reside in the high-density region of the data manifold specified by $\mu_\phi$ through optimizing $\theta$. Thus, the loss effectively creates a trust region defined by the diffusion-based behavior-cloning policy, within which the one-step policy $\pi_\theta$ can move freely. If the generated action deviates significantly from this trust region, it will be heavily penalized.

**Remark 1.** *For any given $s$, assuming that our training set consists of a finite number of samples $\{a_0^1, \ldots, a_0^n\}$, this implies that $p(x|s)$ is represented by a mixture of Dirac delta distributions:*

$$p(x|s) = \frac{1}{n} \sum_{i=1}^{n} \delta(x - a_0^i)$$

*This indicates that all actions $a_0^i$ appearing in the training set have a uniform probability mass. Therefore, the generated action $a_\theta$ can be any one of the actions in $\{a_0^1, \ldots, a_0^n\}$ to minimize $\mathcal{L}_{TR}(\theta)$ in Equation 4, since all of them are modes of the data distribution.*

**Remark 2.** *This loss is also closely connected with Diffusion-GAN [Wang et al., 2022b] and EB-GAN [Zhao et al., 2016], where the discriminator loss is considered as:*

$$D(a_\theta|s) = \|Dec(Enc(a_\theta)|s) - a_\theta\|_2^2$$

*In our model, the process of adding noise, $\alpha_t a_\theta + \sigma_t \epsilon$, functions as an encoder, and $\mu_\phi(\cdot|s)$ acts as a decoder. Thus, this loss can also be considered as a discriminator loss, which determines whether the generated action $a_\theta$ resembles the training dataset.*

**Remark 3.** *By Theorem 1, the trust region can be defined using the conditional log-likelihood. Specifically, for a given state $s$, the trust region for an action is defined as the set $\{a \mid \log p(a \mid s) \geq threshold\}$, where the conditional log-likelihood is approximated by the diffusion loss. The threshold can be adjusted by tuning the hyperparameter $\alpha$ during the optimization of the final loss (Eq. 5).*

This approach makes the generated action $a_\theta$ appear similar to in-sample actions and penalizes those that differ, thereby effectuating behavior regularization. Thus, a visualization of the toy examples (Figure 1) can help better understand how this loss behaves. The generated action $a_\theta$ will incur a small diffusion loss when it resembles a true in-sample action and a high diffusion loss if it deviates significantly from the true in-sample action.

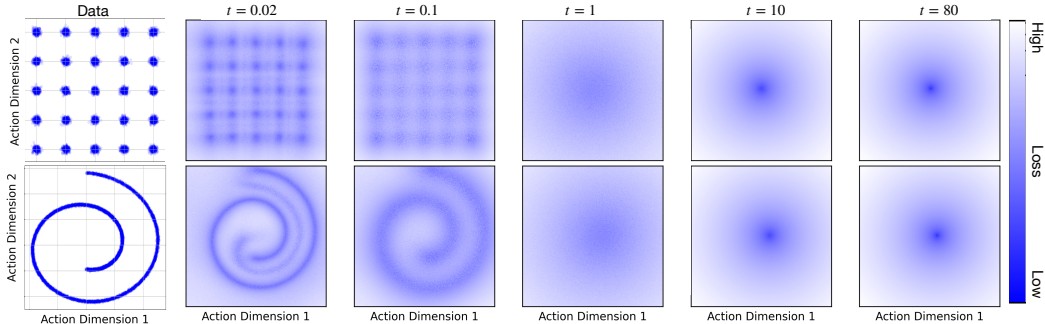

Figure 1: Diffusion trust region loss. The first column shows how the training behavior dataset looks. Columns 2-6 display the diffusion trust region loss on different actions with varying magnitudes of Gaussian noise. We can observe that the trust regions captured by the diffusion model at a given $t$ are consistent with the high-density regions of the noisy data at that specific $t$. For example, when $t$ is small, the diffusion loss is minimal where the true action lies, and high in all other locations.

## 2.4 Diffusion Trusted Q-Learning

We motivate our final algorithm from DQL [Wang et al., 2022a], which utilizes a diffusion model as an expressive policy to facilitate accurate policy regularization, ensuring that exploration remains within a safe region. Q-learning is implemented by maximizing the Q-value function at actions sampled from

the diffusion policy. However, sampling actions from diffusion models can be time-consuming, and computing gradients of the Q-value function while backpropagating through all diffusion timesteps may result in a vanishing gradient problem, especially when the number of timesteps is substantial.

Building on this, we introduce a dual-policy approach, Diffusion Trusted Q-Learning (DTQL): a diffusion policy for pure behavior cloning and a one-step policy for actual depolyment. We bridge the two policies through our newly introduced diffusion trust region loss, detailed in Section 2.3. The diffusion policy ensures that behavior cloning remains expressive, while the trust region loss enables the one-step policy to explore freely and seek modes within the region designated by the diffusion policy. The trust region loss is optimized efficiently through each diffusion timestep without requiring the inference of the diffusion policy. DTQL not only maintains an expressive exploration region but also facilitates efficient optimization. We further discuss the mode-seeking behavior of the diffusion trust region loss in Section 3. Next, we delve into the specifics of our algorithm.

**Policy Learning.** Diffusion inference is not required during training or evaluation in our algorithm; therefore, we utilize an unlimited number of timesteps and construct the diffusion policy $\mu_\phi$ in a continuous-time setting, based on the schedule outlined in EDM [Karras et al., 2022]. Further details are provided in Section 4.3. The diffusion policy $\mu_\phi$ can be efficiently optimized by minimizing $\mathcal{L}(\phi)$ as described in Equation (2). Furthermore, we can instantiate one typical one-step policy $\pi_\theta(\boldsymbol{a}|\boldsymbol{s})$ in two cases, Gaussian $\pi_\theta(\boldsymbol{a}|\boldsymbol{s}) = \mathcal{N}(\mu_\theta(\boldsymbol{s}), \sigma_\theta(\boldsymbol{s}))$ or Implicit $\boldsymbol{a}_\theta = \pi_\theta(\boldsymbol{s}, \varepsilon), \varepsilon \sim \mathcal{N}(0, \boldsymbol{I})$. Then, we optimize $\pi_\theta$ by minimizing the introduced diffusion trust region loss and typical Q-value function maximization, as follows.

$$\mathcal{L}_\pi(\theta) = \alpha \cdot \mathcal{L}_{\text{TR}}(\theta) - \mathbb{E}_{\boldsymbol{s}\sim\mathcal{D}, \boldsymbol{a}_\theta\sim\pi_\theta(\boldsymbol{a}|\boldsymbol{s})}[Q_\eta(\boldsymbol{s}, \boldsymbol{a}_\theta)], \tag{5}$$

where $\mathcal{L}_{\text{TR}}(\theta)$ serves primarily as a behavior-regularization term, and maximizing the Q-value function enables the model to preferentially sample actions associated with higher values. Here we use the double Q-learning trick [Hasselt, 2010] where $Q_\eta(\boldsymbol{s}, \boldsymbol{a}_\theta) = \min(Q_{\eta_1}(\boldsymbol{s}, \boldsymbol{a}_\theta), Q_{\eta_2}(\boldsymbol{s}, \boldsymbol{a}_\theta))$. If a Gaussian policy is used, an additional negative log likelihood (NLL) term, $-\mathbb{E}_{\boldsymbol{s}, \boldsymbol{a}\sim\mathcal{D}}[\log \pi_\theta(\boldsymbol{a}|\boldsymbol{s})]$, should be introduced to preserve the policy's entropy and encourage exploration during training. This aspect is particularly crucial for diverse and sparse reward RL tasks. The empirical results of the NLL term will be discussed in Section 4.4.

**Q-Learning.** We utilize Implicit Q-Learning (IQL) to train a Q function by maintaining two Q-functions $(Q_{\eta_1}, Q_{\eta_2})$ and one value function $V_\psi$, following the methodology outlined in IQL [Kostrikov et al., 2021].

The loss function for the value function $V_\psi$ is defined as:

$$\mathcal{L}_V(\psi) = \mathbb{E}_{(\boldsymbol{s}, \boldsymbol{a}\sim\mathcal{D})} \left[ L_2^\tau \left( \min(Q_{\eta_1'}(\boldsymbol{s}, \boldsymbol{a}), Q_{\eta_2'}(\boldsymbol{s}, \boldsymbol{a})) - V_\psi(\boldsymbol{s}) \right) \right], \tag{6}$$

where $\tau$ is a quantile in $[0, 1]$, and $L_2^\tau(u) = |\tau - \mathbf{1}(u < 0)|u^2$. When $\tau = 0.5$, $L_2^\tau$ simplifies to the $L_2$ loss. When $\tau > 0.5$, $L_\psi$ encourages the learning of the $\tau$ quantile values of $Q$.

The loss function for updating the Q-functions, $Q_{\eta_i}$, is given by:

$$\mathcal{L}_Q(\eta_i) = \mathbb{E}_{(\boldsymbol{s}, \boldsymbol{a}, \boldsymbol{s}'\sim\mathcal{D})} \left[ ||r(\boldsymbol{s}, \boldsymbol{a}) + \gamma * V_\psi(\boldsymbol{s}') - Q_{\eta_i}(\boldsymbol{s}, \boldsymbol{a})||^2 \right], \tag{7}$$

where $\gamma$ denotes the discount factor. This setup aims to minimize the error between the predicted Q-values and the target values derived from the value function $V_\psi$ and the rewards. We summarize our algorithm in Algorithm 1.

## 3 Comparison of Different Mode-Seeking Behavior Regularizations

Another approach to accelerate training and inference in diffusion-based policy learning involves utilizing distillation techniques. Methods such as SDS [Poole et al., 2022], VSD [Wang et al., 2024], Diff-Instruct [Luo et al., 2024], and DMD [Yin et al., 2023] illustrate this strategy. These papers share a common theme: using a trained diffusion model alongside another diffusion network to minimize the KL divergence between the two models. In our experimental setup, this strategy is employed for behavior regularization by

$$\mathcal{L}_{\text{KL}}(\theta) = D_{\text{KL}}[\pi_\theta(\cdot|\boldsymbol{s})||\mu_\phi(\cdot|\boldsymbol{s})] = \mathbb{E}_{\varepsilon\sim\mathcal{N}(0,\boldsymbol{I}), \boldsymbol{s}\sim\mathcal{D}, \pi_\theta(\boldsymbol{s},\varepsilon)} \left[ \log \frac{p_{\text{fake}}(\boldsymbol{a}_\theta|\boldsymbol{s})}{p_{\text{real}}(\boldsymbol{a}_\theta|\boldsymbol{s})} \right] \tag{8}$$

where $\pi_\theta(\boldsymbol{s}, \varepsilon)$ is instantiated as a one-step implicit policy.

**Algorithm 1** Diffusion Trusted Q-Llearning

---

Initialize policy network $\pi_\theta$, $\mu_\phi$, critic networks $Q_{\eta_1}$ and $Q_{\eta_2}$, and target networks $Q_{\eta_1'}$ and $Q_{\eta_2'}$, value function $V_\psi$
**for** each iteration **do**
    Sample transition mini-batch $\mathcal{B} = \{(\boldsymbol{s}_t, \boldsymbol{a}_t, r_t, \boldsymbol{s}_{t+1})\} \sim \mathcal{D}$ .
    1. Q-value function learning:  Update $Q_{\eta_1}$, $Q_{\eta_2}$ and $V_\psi$ by $\mathcal{L}_Q$ and $\mathcal{L}_V$ (Eqs. 6 and 7).
    2. Diffusion Policy learning:  Update $\mu_\phi$ by $\mathcal{L}(\phi)$ (Eq. 2).
    3. Diffusion Trust Region Policy learning:  $\boldsymbol{a}_\theta \sim \pi_\theta(\boldsymbol{a}|\boldsymbol{s})$, Update $\pi_\theta$ by $\mathcal{L}_\pi(\theta)$ (Eq. 5).
    4. Update target networks:  $\eta_i' = \rho\eta_i' + (1-\rho)\eta_i$ for $i = \{1, 2\}$.
**end for**

---

As we do not have access to the log densities of the fake and true conditional distributions of actions, the loss itself cannot be calculated directly. However, we are able to compute the gradients. The gradient of $\log p_{\text{real}}(\boldsymbol{a}_\theta|\boldsymbol{s})$ can be estimated by the diffusion model $\mu_\phi(\cdot|\boldsymbol{s})$, and the gradient of $\log p_{\text{fake}}(\boldsymbol{a}_\theta|\boldsymbol{s})$ can also be estimated by a diffusion model trained from fake action data $\boldsymbol{a}_\theta$. For more details, please refer to Appendix C.

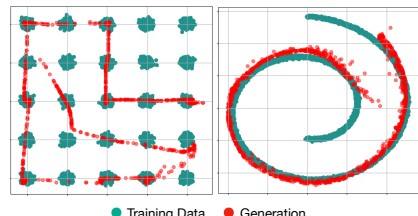

● Training Data  ● Generation

KL divergence is employed in this context with the goal of capturing **multiple modalities** of the data distribution. We evaluated this loss function using a 2D toy task to gain a deeper understanding of its capability to capture multiple modalities of the dataset, as illustrated in Figure 2.

Figure 2: Green points represent the datasets we trained on. Red points are generated by $\pi_\theta$, trained using $\mathcal{L}_{\text{KL}}$. This demonstrates that the KL loss encourages the generation process to cover multiple modalities of the dataset.

We further investigate the differences between our trust region loss, $\mathcal{L}_{\text{TR}}$, and the KL-based behavior distillation loss within the context of policy improvement. As illustrated in Figure 1, $\mathcal{L}_{\text{TR}}$ ensures that the generated action $\boldsymbol{a}_\theta$ remains within the action manifold of the in-sample dataset. Coupled with the gradient of the Q-function, this allows actions to move freely within the in-sample data manifold while gravitating toward high-reward regions, which correspond to the **single modality** present in the dataset.

Conversely, $\mathcal{L}_{\text{KL}}(\theta)$ seeks to align the distribution of $\pi_\theta(\cdot|\boldsymbol{s})$ with that of $\mu_\phi(\cdot|\boldsymbol{s})$, thereby encouraging coverage of **multiple modalities**, unlike $\mathcal{L}_{\text{TR}}$. Covering a wide range of modalities is particularly beneficial in image generation, where diversity among generated images is essential. However, this characteristic is less advantageous in reinforcement learning (RL) contexts, where typically a single, highest-reward action is optimal for a given state. Additionally, maximizing the Q function often results in a more deterministic policy by favoring the highest-reward paths, potentially discarding alternative actions. From this perspective, $\mathcal{L}_{\text{TR}}$ demonstrates a stronger mode-seeking capability compared to $\mathcal{L}_{\text{KL}}$.

To visualize how these two different behavior losses work with policy improvement, we use 2D bandit scenarios. We designed a scenario shown in Figure 3; for additional settings, please refer to Appendix F.1. In the designed 25 Gaussian setting, all four corners have the same high reward. $\mathcal{L}_{\text{TR}}$ encourages the policy to randomly select one high reward mode without promoting covering all of them. In contrast, $\mathcal{L}_{\text{KL}}$ tries to cover all high-density and high-reward regions and, as a byproduct, introduces artifacts that appear as data connecting these high-density regions. This could partially be due to the smoothness constraint of neural networks. The same situation occurs in a Swiss roll dataset where the high reward region is the center of the data; $\mathcal{L}_{\text{TR}}$ adheres closely to the high reward region, while $\mathcal{L}_{\text{KL}}$ includes some suboptimal reward regions.

In addition to testing on 2D bandit scenarios, we also evaluated the performance of two losses on the Mujoco Gym Medium task. Consistent with our previous findings, the behavior-regularization loss $\mathcal{L}_{\text{TR}}(\theta)$ consistently outperformed $\mathcal{L}_{\text{KL}}(\theta)$ in terms of achieving higher rewards. The results are presented in Table 5, and the training curves are depicted in Figure 8 in Appendix F.2.

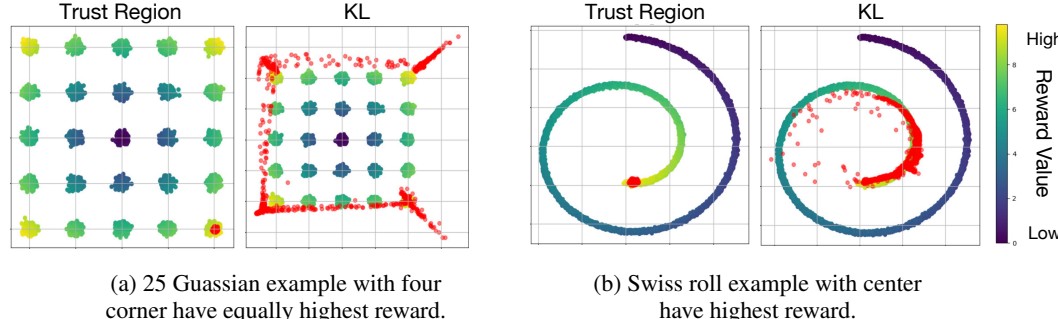

(a) 25 Guassian example with four corner have equally highest reward.

(b) Swiss roll example with center have highest reward.

Figure 3: We tested the differential impact of $\mathcal{L}_{\text{TR}}$ and $\mathcal{L}_{\text{KL}}$ on behavior regularization, using a trained Q-function for policy improvement. Red points represent actions generated from the one-step policy $\pi_\theta$.

**Connection and Difference with SDS and SRPO**    SDS was first proposed in DreamFusion [Poole et al., 2022] for 3D generation, using the gradient of the loss form (adopted in our setting):

$$\nabla_\theta \mathcal{L}_{\text{SDS}} = \mathbb{E}_{t,\boldsymbol{s},\boldsymbol{\varepsilon}} \left[ w(t)(\varepsilon_\phi(\boldsymbol{z}_t, t|\boldsymbol{s}) - \boldsymbol{\varepsilon}) \frac{\partial \boldsymbol{z}_t}{\partial \theta} \right] \tag{9}$$

where $\boldsymbol{z}_t = \alpha_t \boldsymbol{a}_\theta + \sigma_t \boldsymbol{\varepsilon}$ and $\varepsilon_\phi$ is the noise-prediction diffusion model. This loss is utilized by SRPO [Chen et al., 2023] in offline RL.

Considering the gradient of $\mathcal{L}_{\text{TR}}(\theta)$ in Equation 4, and acknowledging the equivalence between noise-prediction and data-prediction diffusion models with only a modification in the weight function $w(t)$, we can reformulate the loss in noise-prediction form by:

$$\mathcal{L}_{\text{TR}}(\theta) = \mathbb{E}_{t,\boldsymbol{s},\boldsymbol{\varepsilon}}[w'(t)\|\varepsilon_\phi(\boldsymbol{z}_t, t|\boldsymbol{s}) - \boldsymbol{\varepsilon}\|_2^2] \tag{10}$$

$$\nabla_\theta \mathcal{L}_{\text{TR}}(\theta) = 2\mathbb{E}_{t,\boldsymbol{s},\boldsymbol{\varepsilon}} \left[ w'(t)(\varepsilon_\phi(\boldsymbol{z}_t, t|\boldsymbol{s}) - \boldsymbol{\varepsilon}) \frac{\partial \varepsilon_\phi(\boldsymbol{z}_t, t|\boldsymbol{s})}{\partial \boldsymbol{z}_t} \frac{\partial \boldsymbol{z}_t}{\partial \theta} \right] \tag{11}$$

The primary distinction between the gradient of our method, as shown in Equation 11, and that of SDS/SRPO, detailed in Equation 9, lies in the inclusion of a Jacobian term, $\frac{\partial \epsilon_\phi(\boldsymbol{z}_t, t|\boldsymbol{s})}{\partial \boldsymbol{z}_t}$. This Jacobian term, identified as the score gradient in SiD by Zhou et al. [2024], is notably absent from most theoretical discussions and was deliberately omitted in previous works, with DreamFusion [Poole et al., 2022] and SiD being the sole exceptions.

DreamFusion reported that the gradient depicted in Equation 11 fails to produce realistic 3D samples. Similarly, SiD observed its inadequacy in generating realistic images. These findings align with our Theorem 1, which demonstrates that this gradient primarily targets the mode and does not sufficiently account for diversity— an essential factor in both 3D and image generation.

In high-dimensional generative models, modes often differ significantly from typical image samples, as discussed by Nalisnick et al. [2018]. DreamFusion observed that the gradient from Equation 9, which is based on a KL loss, effectively promotes diversity. However, while diversity is crucial in image and 3D generation, it might be of lesser importance in offline RL. Consequently, SRPO's use of the SDS gradient, which is tailored for diverse generation, may result in suboptimal performance compared to our diffusion trust region loss. This assertion is supported by empirical results on the D4RL datasets, as discussed in Section 4.1.

## 4    Experiments

In this section, we evaluate our method using the popular D4RL benchmark [Fu et al., 2020]. We further compare our training and inference efficiency against other baseline methods. Additionally, an ablation study on the negative log likelihood (NLL) term and one-step policy choice is presented. Details regarding the training of the diffusion model and its structural components are also discussed.

Table 1: The performance of Our methods and SOTA baselines on D4RL Gym, AntMaze, Adroit, and Kitchen tasks. Results for our methods correspond to the mean and standard errors of normalized scores over 50 random rollouts (5 independently trained models and 10 trajectories per model) for Gym tasks, which generally exhibit low variance in performance, and over 500 random rollouts (5 independently trained models and 100 trajectories per model) for the other tasks. Our method outperforms all prior methods by a clear margin on most of domains. The normalized scores is recorded by the end of training phase. Numbers within 5 % of the maximum in every individual task are highlighted.

| Gym | BC | Onestep RL | TD3+BC | DT | CQL | IQL | DQL | IDQL | SRPO | Ours |
|---|---|---|---|---|---|---|---|---|---|---|
| halfcheetah-medium-v2 | 42.6 | 48.4 | 48.3 | 42.6 | 44.0 | 47.4 | 51.1 | 51.0 | **60.4** | $57.9 \pm 0.13$ |
| hopper-medium-v2 | 52.9 | 59.6 | 59.3 | 67.6 | 58.5 | 66.3 | 90.5 | 65.4 | 95.5 | **99.6**±0.87 |
| walker2d-medium-v2 | 75.6 | 81.8 | 83.7 | 74.0 | 72.5 | 78.3 | **87.0** | 82.5 | 84.4 | **89.4**±0.13 |
| halfcheetah-medium-replay-v2 | 36.3 | 38.1 | 44.6 | 36.0 | 45.2 | 44.2 | 47.8 | 45.9 | **51.4** | **50.9**±0.11 |
| hopper-medium-replay-v2 | 18.1 | 97.5 | 60.9 | 82.7 | 95.0 | 94.7 | 101.3 | 92.1 | **101.2** | **100.0**±0.13 |
| walker2d-medium-replay-v2 | 26.0 | 49.5 | 81.8 | 66.6 | 77.2 | 73.9 | **95.5** | 85.1 | 84.6 | 88.5± 2.16 |
| halfcheetah-medium-expert-v2 | 55.2 | 93.4 | 90.7 | 86.8 | 91.6 | 86.7 | **96.8** | 95.9 | 92.2 | $92.7 \pm 0.2$ |
| hopper-medium-expert-v2 | 52.5 | 103.3 | 98.0 | 107.6 | 105.8 | 91.5 | **111.1** | 108.6 | 100.1 | $109.3 \pm 1.49$ |
| walker2d-medium-expert-v2 | 101.9 | **113.0** | 110.1 | 107.1 | 109.4 | 109.6 | 110.1 | 112.7 | **114.0** | $110 \pm 0.07$ |
| **Gym Average** | 51.9 | 76.1 | 75.3 | 74.7 | 77.6 | 77.0 | 88.0 | 82.1 | **87.1** | 88.7 |

| Antmaze | BC | Onestep RL | TD3+BC | DT | CQL | IQL | DQL | IDQL | SRPO | Ours |
|---|---|---|---|---|---|---|---|---|---|---|
| antmaze-umaze-v0 | 54.6 | 64.3 | 78.6 | 59.2 | 74.0 | 87.5 | 93.4 | 94.0 | 90.8 | **94.8**±1.00 |
| antmaze-umaze-diverse-v0 | 45.6 | 60.7 | 71.4 | 53.0 | **84.0** | 62.2 | 66.2 | 80.2 | 59.0 | 78.8±1.83 |
| antmaze-medium-play-v0 | 0.0 | 10.6 | 0.0 | 0.0 | 61.2 | 71.2 | 76.6 | **84.5** | 73.0 | $79.6 \pm 1.8$ |
| antmaze-medium-diverse-v0 | 0.0 | 3.0 | 0.2 | 0.0 | 53.7 | 70.0 | 78.6 | **84.8** | 65.2 | $82.2 \pm 1.71$ |
| antmaze-large-play-v0 | 0.0 | 0.0 | 0.0 | 0.0 | 15.8 | 39.6 | 46.4 | **63.5** | 38.8 | 52.0± 2.23 |
| antmaze-large-diverse-v0 | 0.0 | 0.0 | 0.0 | 0.0 | 14.9 | 47.5 | 56.6 | **67.9** | 33.8 | 54.0 ± 2.23 |
| **Antmaze Average** | 16.7 | 20.9 | 27.3 | 18.7 | 50.6 | 63.0 | 69.6 | **79.1** | 30.1 | 73.6 |

| Adroit Tasks | BC | BCQ | BEAR | BRAC-p | BRAC-v | REM | CQL | IQL | DQL | Ours |
|---|---|---|---|---|---|---|---|---|---|---|
| pen-human-v1 | 25.8 | 68.9 | -1.0 | 8.1 | 0.6 | 5.4 | 35.2 | **71.5** | **72.8** | 64.1±2.97 |
| pen-cloned-v1 | 38.3 | 44.0 | 26.5 | 1.6 | -2.5 | -1.0 | 27.2 | 37.3 | 57.3 | **81.3**± 3.04 |
| **Adroit Average** | 32.1 | 56.5 | 12.8 | 4.9 | -1.0 | 2.2 | 31.2 | 54.4 | 65.1 | **72.7** |

| Kitchen Tasks | BC | BCQ | BEAR | BRAC-p | BRAC-v | AWR | CQL | IQL | DQL | Ours |
|---|---|---|---|---|---|---|---|---|---|---|
| kitchen-complete-v0 | 33.8 | 8.1 | 0.0 | 0.0 | 0.0 | 0.0 | 43.8 | 62.5 | **84.0** | 80.8±1.06 |
| kitchen-partial-v0 | 33.8 | 18.9 | 13.1 | 0.0 | 0.0 | 15.4 | 49.8 | 46.3 | 60.5 | **74.4**±0.25 |
| kitchen-mixed-v0 | 47.5 | 8.1 | 47.2 | 0.0 | 0.0 | 10.6 | 51.0 | 51.0 | **62.6** | **60.2**±0.59 |
| **Kitchen Average** | 38.4 | 11.7 | 20.1 | 0.0 | 0.0 | 8.7 | 48.2 | 53.3 | **69.0** | **71.8** |

**Hyperparameters**   In D4RL benchmarks, for all Antmaze tasks, we incorporate an NLL term, while for other tasks, this term is omitted. Additionally, we adjust the parameter $\alpha$ for different tasks. Details on hyperparameters and implementation are provided in Appendices D and E.

## 4.1   D4RL Performance

In Table 1, we evaluate the D4RL performance of our method against other offline algorithms. Our selected benchmarks include conventional methods such as TD3+BC [Fujimoto and Gu, 2021] and IQL [Kostrikov et al., 2021], along with newer diffusion-based models like Diffusion QL (DQL) [Wang et al., 2022a], IDQL [Hansen-Estruch et al., 2023], and SRPO [Chen et al., 2023].

In the D4RL datasets, our method (DTQL) outperformed all conventional and other diffusion-based offline RL methods, including DQL and SRPO, across all tasks. Moreover, it is 10 times more efficient in inference than DQL and IDQL; and 5 times more efficient in total training wall time compared with IDQL (see Section 4.2).

**Remark 4.** *We would like to highlight that the SRPO method [Chen et al., 2023] reported results on Antmaze using the "-v2" version, which differs from the "-v0" version employed by prior methods such as DQL [Wang et al., 2022a] and IDQL [Hansen-Estruch et al., 2023], to which it was compared. This version discrepancy, not explicitly stated in their paper, is evident upon inspection of SRPO's official codebase [1]. The variation between the -v2" and -v0" datasets significantly impacts algorithm performance. To ensure a fair comparison, we utilize the "-v0" environments consistent with established baselines. We employed the official SRPO code on Antemze-v0 and maintained identical hyperparameters used for Antmaze-v2. Additionally, we conducted experiments with our*

---

[1]Refer to line 7 at `https://github.com/thu-ml/SRPO/blob/main/utils.py`, commit b006412

Table 2: Training and Inference time required for different algorithms in D4RL *antmaze-umaze-v0* tasks. Every single experiment is conducted with the same PyTorch backend and run on a single RTX-A5000 GPU.

| antmaze-umaze-v0 | DQL | IDQL | SRPO | Ours |
|---|---|---|---|---|
| Training time (s per 1k steps) | 24.13 | 17.57 | 24.71 | 21.83 |
| Inference time (s per trajectory) | 3.03 | 3.04 | 0.22 | 0.35 |
| Training epochs | 1000 | 3000 | 1000 | 500 |
| Total training time (hours) | 6.70 | 14.64 | 9.42 | 3.33 |

*algorithm on the Antmaze-v2 environment using the same hyperparameters as in the Antmaze-v0 setup but extended the training epochs, as detailed in Table 6 in Appendix F.*

## 4.2 Computational Efficiency

We further examine the training and inference performance relative to other diffusion-based offline RL methods. An overview of this performance, using *antmaze-umaze-v0* as a benchmark, is presented in Table 2. Our method requires less training time per epoch than DQL and SRPO, yet more than IDQL. However, while IDQL necessitates 3000 epochs, DTQL operates efficiently with only 500 epochs, considerably reducing the overall training duration.

As depicted in Figure 4, the extended training time per epoch for our method results from the requirement to train an additional one-step policy, a step not needed by IDQL. Although SRPO also incorporates a one-step policy, our method achieves greater efficiency in training the diffusion policy. Unlike SRPO, which requires several ResNet blocks for effective performance, our approach utilizes only a 4-layer MLP, further curtailing the training time. Additional details on total training wall time are provided in Appendix F.4.

For inference time, our method performs comparably to SRPO, as both utilize a one-step policy. The slightly higher inference time results from our use of a stochastic policy, which requires resampling after each forward pass of the neural network. Additionally, we employ the stochastic max Q trick, similar to that used in DQL Wang et al. [2022a]. However, our method achieves a tenfold increase in inference speed over DQL and IDQL, which require 5-step iterative denoising to generate actions. All experiments were performed on a server equipped with eight RTX-A5000 GPUs, each with 24GB of memory.

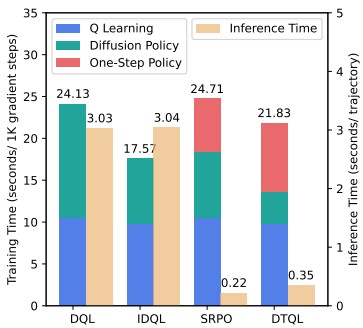

**Remark 5.** *For total training time, SRPO trains 1000 epochs for the one-step policy while training 1500 epochs for the diffusion policy and Q function. DTQL requires 50 epochs of pretraining. Implement details are in Appendix D.*

Figure 4: Training time required for different algorithms in D4RL *antmaze-umaze-v0* tasks. All experiments are conducted with the same PyTorch backend and the same computing hardware setup.

## 4.3 Diffusion Training Schedule

For training the diffusion policy as described in Equation 2 and the diffusion trust region loss in Equation 4, we utilize the diffusion weight and noise schedule outlined in EDM [Karras et al., 2022]. Although EDM does not satisfy the ELBO condition stipulated in Equation 3—a fact established in Kingma and Gao [2024]—we adopted it due to its demonstrated enhancements in perceptual generation quality, as evidenced by metrics such as the Fréchet Inception Distance (FID) and Inception Score in the field of image generation. Kingma and Gao [2024] also attempted to modify the EDM weight schedule to be monotonically increasing, but this did not lead to better FID for image generation. Thus, we retain EDM as our continuous training schedule. For completeness, the details of the EDM schedule are discussed in Appendix B.

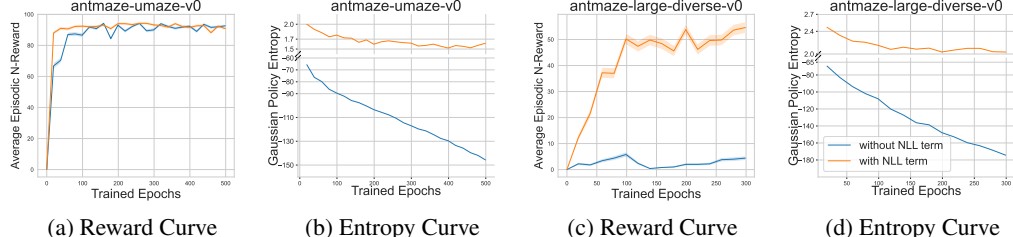

|  (a) Reward Curve | (b) Entropy Curve | (c) Reward Curve | (d) Entropy Curve |

Figure 5: Rewards and Gaussian policy entropy during training are recorded and illustrated in the figures. The blue line represents training without the addition of an NLL term, while the orange line indicates training with the NLL term included.

## 4.4 Ablation Studies

**One-step Policy Choice**   We chose to use a Gaussian policy for all our experiments instead of an implicit or deterministic policy because the Gaussian policy is flexible and provides a convenient way to control entropy when needed. When there is no need to maintain entropy, the Gaussian policy quickly degenerates to a deterministic policy, where the variance approaches zero, as indicated in Figures 5b and 5d.

**Negative Log Likelihood Term**   As mentioned in Section 2.4, we incorporate an NLL term $-\mathbb{E}_{\boldsymbol{s},\boldsymbol{a}\sim\mathcal{D}}[\log \pi(\boldsymbol{a}|\boldsymbol{s})]$ into the loss function in Equation 5 to maintain exploration and policy entropy during training when using a Gaussian policy. We conducted an ablation study to assess its impact on the final rewards and the of the Gaussian policy, taking *antmaze-umaze-v0* and *antmaze-large-diverse-v0* as examples. As observed in Figure 5, for the less complex task *antmaze-umaze-v0*, adding the NLL term does not significantly enhance the final score but does stabilize the training process (see Figure 5a). However, for more complex tasks like *antmaze-large-diverse-v0*, the addition of the NLL term markedly increases the final score. We attribute this improvement to the ability of the NLL term to maintain high entropy during training, thus preserving exploration capabilities, as shown in Figures 5b and 5d.

## 5   Conclusion and Limitation

In this work, we present DTQL, which comprises a diffusion policy for pure behavior cloning and a practical one-step policy. The diffusion policy maintains expressiveness, while the diffusion trust region loss introduced in this paper directs the one-step policy to explore freely and seek modes within the safe region defined by the diffusion policy. This training pipeline eliminates the need for iterative denoising sampling during both training and inference, making it remarkably computationally efficient. Moreover, DTQL achieves state-of-the-art performance across the majority of tasks in the D4RL benchmark. Some limitations of DTQL include the potential for improvement in its benchmark performance. Additionally, some design aspects of the one-step policy could benefit from further investigation. Currently, our experiments are primarily conducted in an offline setting. It would be interesting to explore how this method can be extended to an online setting or adapted to handle more complex inputs, such as images. Additionally, rather than focusing solely on point estimation of rewards, it would be beneficial to estimate the distribution of rewards, as recommended by previous studies in distributional reinforcement learning [Bellemare et al., 2017, Barth-Maron et al., 2018, Yue et al., 2020].

## Acknowledgments

The authors acknowledge the support of NSF-IIS 2212418 and NIH-R37 CA271186.

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

# Supplementary Material: Diffusion Policies Creating a Trust Region for Offline Reinforcement Learning

## A   Related Work

**Expressive Generative Models for Behavior Cloning**   Behavior cloning refers to the task of learning the behavior policy that was used to collect static datasets. Generative models are often employed for behavior cloning due to their expressive power. For instance, EMaQ [Ghasemipour et al., 2021] uses an auto-regressive model for behavior cloning. BCQ [Fujimoto et al., 2019] utilizes a Conditional Variational Autoencoder (VAE), while Florence et al. [2022] employ energy-based models. GAN-Joint [Yang et al., 2022] leverages GANs, and several studies [Wang et al., 2022a, Janner et al., 2022, Pearce et al., 2023] utilize diffusion models for behavior cloning. Diffusion models have demonstrated strong performance due to their ability to capture multimodal distributions. However, they may suffer from increased training and inference times because of the iterative denoising process required for sampling.

**Efficiency Improvement in Diffusion-Based RL Methods.**   Several studies aim to accelerate the training of diffusion models in offline RL settings. One approach involves using specialized diffusion ODE solvers, such as the DDIM solver [Song et al., 2020a] or the DPM-solver [Lu et al., 2022], to speed up iterative sampling. Another strategy is to avoid iterative denoising during training or inference. EDP [Kang et al., 2024] and IDQL [Hansen-Estruch et al., 2023] both focus on avoiding iterative sampling during training. EDP adopts an approximate diffusion sampling scheme to minimize the required sampling steps, although it still requires iterative denoising during inference. IDQL accelerates the training process by only training a behavior cloning policy without denoising sampling. However, it requires iterative sampling during inference by selecting from a batch of candidate generated actions. SRPO [Chen et al., 2023] employs score distillation methods to avoid iterative denoising in both training and inference.

**Distillation Methods.**   Distillation methods for diffusion models have been proposed to enable one-step generation of images or 3D objects. Examples of such methods include SDS [Poole et al., 2022], VSD [Wang et al., 2024], Diff Instruct [Luo et al., 2024], and DMD [Yin et al., 2023]. The core idea of these methods is to minimize the KL divergence between a pre-trained diffusion model and a target one-step generation model. SiD [Zhou et al., 2024] uses a different divergence metric but shares the same goal of mimicking the distribution learned by a pre-trained diffusion model. The distillation strategy can also be applied in the offline RL field to accelerate training and inference. However, directly adopting these methods may result in suboptimal performance.

## B   Diffusion Schedule

This diffusion training schedule is the same for training the behavior-cloning policy in Equation 2 and the diffusion trust region loss in Equation 4.

**Noise Schedule**   We illustrate the EDM diffusion training schedule in our setting. First, we need to define some prespecified parameters: $\sigma_{\text{data}} = 0.5$, $\sigma_{\text{min}} = 0.002$, $\sigma_{\text{max}} = 80$. The noise schedule is defined by $\boldsymbol{a}_t = \alpha_t \boldsymbol{a} + \sigma_t \boldsymbol{\varepsilon}$, where $\boldsymbol{\varepsilon} \sim \mathcal{N}(0, \boldsymbol{I})$. We set $\alpha_t = 1$ and $\sigma_t = t$. The variable $\log(t)$ follows a logistic distribution with location parameter $\log \sigma_{\text{data}}$ and scale parameter $0.5$. The original EDM paper samples $\log(t)$ from $\mathcal{N}(-1.2, 1.2^2)$, but this difference does not significantly affect our algorithm.

**Denoiser**   The denoiser $\mu_\phi$ is defined as:

$$\mu_\phi(\boldsymbol{a}_t, t | \boldsymbol{s}) = c_{\text{skip}}(\sigma)\boldsymbol{a}_t + c_{\text{out}}(\sigma)F_\phi(c_{\text{in}}(\sigma)\boldsymbol{a}_t, c_{\text{noise}}(\sigma)|\boldsymbol{s}),$$

where $\sigma = \sigma_t = t$ and $F_\phi$ represents the raw neural network layer. We also define:

$$c_{\text{skip}}(\sigma) = \frac{\sigma_{\text{data}}^2}{\sigma^2 + \sigma_{\text{data}}^2}, \quad c_{\text{out}}(\sigma) = \frac{\sigma \cdot \sigma_{\text{data}}}{\sqrt{\sigma^2 + \sigma_{\text{data}}^2}},$$

$$c_{\text{in}}(\sigma) = \frac{1}{\sigma^2 + \sigma_{\text{data}}^2}, \quad c_{\text{noise}}(\sigma) = \frac{1}{4}\log(\sigma).$$

**Weight Schedule**   The final loss is given by:

$$\mathbb{E}_{\sigma,\boldsymbol{a},\boldsymbol{s},\boldsymbol{\varepsilon}}\left[\lambda(\sigma)c_{\text{out}}^2(\sigma)\left\|F_\phi(c_{\text{in}}(\sigma)\cdot(\boldsymbol{a}+\boldsymbol{\varepsilon}), c_{\text{noise}}(\sigma)|\boldsymbol{s}) - \frac{1}{c_{\text{out}}(\sigma)}\left(\boldsymbol{a} - c_{\text{skip}}(\sigma)\cdot(\boldsymbol{a}+\boldsymbol{\varepsilon})\right)\right\|_2^2\right],$$

where $\lambda(\sigma) = \frac{1}{c_{\text{out}}^2(\sigma)}$.

## C   Details in KL Behavior Regularization

Here we introduce how we implement KL divergence regularization. The idea is similar to previous KL-based distillation methods [Wang et al., 2024, Luo et al., 2024, Yin et al., 2023], but adapted to our setting. Our loss function is defined as:

$$\mathcal{L}_{\text{KL}}(\theta) = D_{\text{KL}}[\pi_\theta(\cdot|\boldsymbol{s})||\mu_\phi(\cdot|\boldsymbol{s})] = \mathbb{E}_{\boldsymbol{\varepsilon}\sim\mathcal{N}(0,\boldsymbol{I}),\boldsymbol{s}\sim\mathcal{D},\pi_\theta(\boldsymbol{s},\boldsymbol{\varepsilon})}\left[\log\frac{p_{\text{fake}}(\boldsymbol{a}_\theta|\boldsymbol{s})}{p_{\text{real}}(\boldsymbol{a}_\theta|\boldsymbol{s})}\right] \tag{12}$$

The gradient of $\mathcal{L}_{\text{KL}}(\theta)$ is given by:

$$\nabla_\theta\mathcal{L}_{\text{KL}}(\theta) = \mathbb{E}_{\boldsymbol{\varepsilon},\boldsymbol{s},\boldsymbol{a}_\theta=\pi_\theta(\boldsymbol{s},\boldsymbol{\varepsilon})}\left[\left(s_{\text{fake}}(\boldsymbol{a}_\theta|\boldsymbol{s}) - s_{\text{real}}(\boldsymbol{a}_\theta|\boldsymbol{s})\right)\nabla_\theta\pi_\theta\right]$$

where $s_{\text{real}}(\boldsymbol{a}_\theta|\boldsymbol{s}) = \nabla_{\boldsymbol{a}_\theta}\log p_{\text{real}}(\boldsymbol{a}_\theta|\boldsymbol{s})$ and $s_{\text{fake}}(\boldsymbol{a}_\theta|\boldsymbol{s}) = \nabla_{\boldsymbol{a}_{theta}}\log p_{\text{fake}}(\boldsymbol{a}_\theta|\boldsymbol{s})$. By using the Score-ODE given in [Song et al., 2020b], we can estimate $s_{\text{real}}(\boldsymbol{a}_\theta|\boldsymbol{s})$ and $s_{\text{fake}}(\boldsymbol{a}_\theta|\boldsymbol{s})$ with a diffusion model. Let $\boldsymbol{a}_{\theta,t} = \alpha_t\boldsymbol{a}_\theta + \sigma_t\boldsymbol{\varepsilon}$, the real score can be estimated by:

$$s_{\text{real}}(\boldsymbol{a}_{\theta,t}, t|\boldsymbol{s}) = -\frac{\boldsymbol{a}_{\theta,t} - \alpha_t\mu_\phi(\boldsymbol{a}_{\theta,t}, t|\boldsymbol{s})}{\sigma_t^2}$$

where $\mu_\phi$ is the pre-trained diffusion behavior cloning model that learns the true data distribution.

Similarly, we can estimate the fake score by:

$$s_{\text{fake}}(\boldsymbol{a}_{\theta,t}, t|\boldsymbol{s}) = -\frac{\boldsymbol{a}_{\theta,t} - \alpha_t\mu_\xi(\boldsymbol{a}_{\theta,t}, t|\boldsymbol{s})}{\sigma_t^2}$$

where $\mu_\xi$ is trained using fake data:

$$\mathcal{L}(\xi) = \|\mu_\xi(\boldsymbol{a}_{\theta,t}, t|\boldsymbol{s}) - \boldsymbol{a}_\theta\|_2^2$$

which is trained with generated fake action data.

Thus, the gradient of $\mathcal{L}_{\text{KL}}(\theta)$ can be expressed as:

$$\nabla_\theta\mathcal{L}_{\text{KL}}(\theta) = \mathbb{E}_{\boldsymbol{\varepsilon},\boldsymbol{s},\boldsymbol{a}_\theta,\boldsymbol{a}_{\theta,t}}\left[w_t\alpha_t\left(s_{\text{fake}}(\boldsymbol{a}_{\theta,t}, t|\boldsymbol{s}) - s_{\text{real}}(\boldsymbol{a}_{\theta,t}, t|\boldsymbol{s})\right)\nabla_\theta\pi_\theta\right]$$

where $w_t = \frac{\sigma_t^2}{\alpha_t}\frac{A}{\|\mu_\phi(\boldsymbol{a}_{\theta,t}, t) - \boldsymbol{a}_\theta\|_1}$ and $A$ is the dimension of the action space.

The algorithm for KL regularization is shown below:

## D   Implementation Details

**Diffusion Policy**   We build our policy as an MLP-based conditional diffusion model. The model itself is an action prediction model. We model $\mu_\phi$ and $\mu_\xi$ as 4-layer MLPs with Mish activations, using 256 hidden units for all networks. The input to $\mu_\phi$ and $\mu_\xi$ is the concatenation of the noisy action vector, the current state vector, and the sinusoidal positional embedding of timestep $t$. The output of $\mu_\phi$ and $\mu_\xi$ is the predicted action at diffusion timestep $t$.

**Algorithm 2** KL Regularization

---

Initialize policy network $\pi_\theta$, $\mu_\phi$, $\mu_\xi$
**for** each iteration **do**
    Sample transition mini-batch $\mathcal{B} = \{(\boldsymbol{s}_t, \boldsymbol{a}_t, r_t, \boldsymbol{s}_{t+1})\} \sim \mathcal{D}$
    Diffusion Policy Learning: Update $\mu_\phi$ by $\mathcal{L}(\phi)$
**end for**
Initialize policy and fake score network: $\theta \leftarrow \phi$, $\xi \leftarrow \phi$
**for** each iteration **do**
    Sample transition mini-batch $\mathcal{B} = \{(\boldsymbol{s}_t, \boldsymbol{a}_t, r_t, \boldsymbol{s}_{t+1})\} \sim \mathcal{D}$, generate $\boldsymbol{a}_\theta$
    Random timestep and add noise: Choose $t$, $\boldsymbol{a}_{\theta_t} = \alpha_t \boldsymbol{a}_\theta + \sigma_t \varepsilon$
    with_no_grad():
        $pred\_fake\_action = \mu_\xi(\boldsymbol{a}_{\theta_t}, t|\boldsymbol{s})$
        $pred\_real\_action = \mu_\phi(\boldsymbol{a}_{\theta_t}, t|\boldsymbol{s})$
    $weighting\_factor = \text{abs}(\boldsymbol{a}_\theta - pred\_real\_action).\text{mean(keepdim=True)}$
    $grad = \frac{pred\_fake\_action - pred\_real\_action}{weighting\_factor}$
    $loss = 0.5 \times \text{mse\_loss}(\boldsymbol{a}_\theta, \text{stopgrad}(\boldsymbol{a}_\theta - grad))$
    Update $\pi_\theta$ by $loss$
    Diffusion Fake Policy Learning: Update $\mu_\xi$ by $\mathcal{L}(\xi)$
**end for**

---

**Q and V Networks** We build two Q networks and a V network with the same MLP setting as our diffusion policy. Each network comprises 4-layer MLPs with Mish activations and 256 hidden units.

**Stochastic Max Q Trick** Similar to DQL Wang et al. [2022a], during inference, we generate $N$ candidate actions and then randomly select an action according to $\exp(Q(\boldsymbol{a}, \boldsymbol{s}))$. Here, $N$ is fixed at 1024 and remains unchanged across different tasks.

**One-Step Policy** We build a Gaussian policy using 3-layer MLPs with ReLU activations, utilizing 256 hidden units. After sampling an action, we apply a tanh activation to ensure the action lies between $[-1, 1]$. If an implicit policy is instantiated, its structure is the same as that of the diffusion policy.

**Pretrain** In our implementation, we pretrain the diffusion policy $\mu_\phi$ and the Q function $Q_\eta$ for 50 epochs to ensure they can better guide $\pi_\theta$. Then, $\mu_\phi$, $Q_\eta$, and $\pi_\theta$ are concurrently trained for the epochs specified in Table 4. We found that introducing a pretrain schedule does not significantly influence the final performance. Our ablation study on the Gym Medium Task revealed that while pretraining yields slightly better results, the final rewards are largely similar. Therefore, we maintain a 50-epoch pretrain for all our tasks. The results are shown in Table 3.

Table 3: The performance with and without pretraining on D4RL Gym tasks.

| Environment | Pretrain | No Pretrain |
|---|---|---|
| halfcheetah-medium-v2 | 57.9 | 57.5 |
| hopper-medium-v2 | 99.6 | 87.6 |
| walker2d-medium-v2 | 89.4 | 88.7 |

# E   Hyperparamaters

Table 4: Hyperparameters for D4RL benchmarks. One epoch represents 1k steps, and the optimizer used is Adam.

| Gym | $\alpha$ | $\tau$ | NLL Term | Pretrain Epochs | Training Epochs | Learning Rate | Lr decay |
|---|---|---|---|---|---|---|---|
| halfcheetah-medium-v2 | 1 | 0.7 | False | 50 | 1000 | $3 \times 10^{-4}$ | False |
| halfcheetah-medium-replay-v2 | 5 | 0.7 | False | 50 | 1000 | $3 \times 10^{-4}$ | False |
| halfcheetah-medium-expert-v2 | 50 | 0.7 | False | 50 | 1000 | $3 \times 10^{-4}$ | False |
| hopper-medium-v2 | 5 | 0.7 | False | 50 | 1000 | $1 \times 10^{-4}$ | True |
| hopper-medium-replay-v2 | 5 | 0.7 | False | 50 | 1000 | $3 \times 10^{-4}$ | False |
| hopper-medium-expert-v2 | 20 | 0.7 | False | 50 | 1000 | $3 \times 10^{-4}$ | False |
| walker2d-medium-v2 | 5 | 0.7 | False | 50 | 1000 | $3 \times 10^{-4}$ | True |
| walker2d-medium-replay-v2 | 5 | 0.7 | False | 50 | 1000 | $3 \times 10^{-4}$ | True |
| walker2d-medium-expert-v2 | 5 | 0.7 | False | 50 | 1000 | $3 \times 10^{-4}$ | True |
| antmaze-umaze-v0 | 1 | 0.9 | True | 50 | 500 | $3 \times 10^{-4}$ | False |
| antmaze-umaze-diverse-v0 | 1 | 0.9 | True | 50 | 500 | $3 \times 10^{-5}$ | True |
| antmaze-medium-play-v0 | 1 | 0.9 | True | 50 | 400 | $3 \times 10^{-4}$ | False |
| antmaze-medium-diverse-v0 | 1 | 0.9 | True | 50 | 400 | $3 \times 10^{-4}$ | False |
| antmaze-large-play-v0 | 1 | 0.9 | True | 50 | 350 | $3 \times 10^{-4}$ | False |
| antmaze-large-diverse-v0 | 0.5 | 0.9 | True | 50 | 300 | $3 \times 10^{-4}$ | False |
| antmaze-umaze-v2 | 1 | 0.9 | True | 50 | 500 | $3 \times 10^{-4}$ | False |
| antmaze-umaze-diverse-v2 | 1 | 0.9 | True | 50 | 500 | $3 \times 10^{-5}$ | True |
| antmaze-medium-play-v2 | 1 | 0.9 | True | 50 | 500 | $3 \times 10^{-4}$ | False |
| antmaze-medium-diverse-v2 | 1 | 0.9 | True | 50 | 500 | $3 \times 10^{-4}$ | False |
| antmaze-large-play-v2 | 1 | 0.9 | True | 50 | 500 | $3 \times 10^{-4}$ | False |
| antmaze-large-diverse-v2 | 0.5 | 0.9 | True | 50 | 500 | $3 \times 10^{-4}$ | False |
| pen-human-v1 | 1500 | 0.9 | False | 50 | 300 | $3 \times 10^{-5}$ | True |
| pen-cloned-v1 | 1500 | 0.7 | False | 50 | 200 | $1 \times 10^{-5}$ | False |
| kitchen-complete-v0 | 200 | 0.7 | False | 50 | 500 | $1 \times 10^{-4}$ | True |
| kitchen-partial-v0 | 100 | 0.7 | False | 50 | 1000 | $1 \times 10^{-4}$ | True |
| kitchen-mixed-v0 | 200 | 0.7 | False | 50 | 500 | $3 \times 10^{-4}$ | True |

# F    Additional Experiments

## F.1    Complete 2D Toy Experiments

We also conducted some 2D bandit experiments with different reward scenarios. In Figure 6, red points are generated by the one-step policy $\pi_\theta$.

In the first column, where the four corners have the same high reward, $\mathcal{L}_{KL}$ tends to encourage exploration of all these high-reward regions, resulting in some suboptimal reward actions. In contrast, $\mathcal{L}_{TR}$ generates actions that randomly select one of the high-reward regions, thereby avoiding suboptimal actions. The same situation occurs in the fourth and fifth columns of Figure 6, where $\mathcal{L}_{KL}$ covers some suboptimal regions while $\mathcal{L}_{TR}$ adheres closely to the highest reward regions.

However, when the data have only one mode with the highest reward, such as in the second and third columns of Figure 6, both $\mathcal{L}_{KL}$ and $\mathcal{L}_{TR}$ guide the policy to generate high-reward actions.

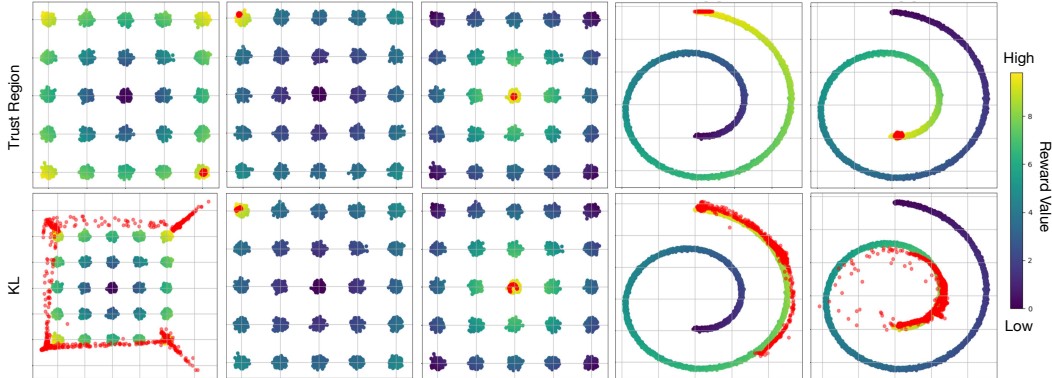

Figure 6: 2D Bandit toy examples, where the behavior regularization is conducted by $\mathcal{L}_{TR}$ and $\mathcal{L}_{KL}$ in different behavior data and reward scenarios. The first row uses behavior regularization by $\mathcal{L}_{TR}$, and the second row uses $\mathcal{L}_{KL}$. Yellow indicates the highest reward, and dark blue indicates the lowest reward.

## F.2    Comparison with KL behavior Regularization in Gym Tasks

In addition to testing on 2D bandit scenarios, we also evaluated the performance of two losses $\mathcal{L}_{KL}$ and $\mathcal{L}_{TR}$ on the Mujoco Gym Medium task. The behavior regularization loss $\mathcal{L}_{TR}(\theta)$ consistently outperformed $\mathcal{L}_{KL}(\theta)$ in terms of achieving higher rewards. The results are presented in Table 5, and the training curves are depicted in Figure 8.

Table 5: The performance of $\mathcal{L}_{TR}(\theta)$ and $\mathcal{L}_{KL}(\theta)$ on D4RL Gym tasks. Results correspond to the mean of normalized scores over 50 random rollouts (5 independently trained models and 10 trajectories per model).

| Environment | $\mathcal{L}_{TR}(\theta)$ | $\mathcal{L}_{KL}(\theta)$ |
|---|---|---|
| halfcheetah-medium-v2 | **57.9** | 24.1 |
| hopper-medium-v2 | **99.6** | 15.0 |
| walker2d-medium-v2 | **89.4** | 3.4 |

## F.3    Comparison with SRPO on Antmaze-v2 Datasets

Since SRPO uses Antmaze-v2 for their D4RL benchmarks, we also conducted experiments on Antmaze-v2 using our algorithm, with the same hyperparameters as those used in Antmaze-v0 but with more training epochs. Hyperparameters details can be found in Table 4. The results for Antmaze-v2 from SRPO are taken directly from their paper.

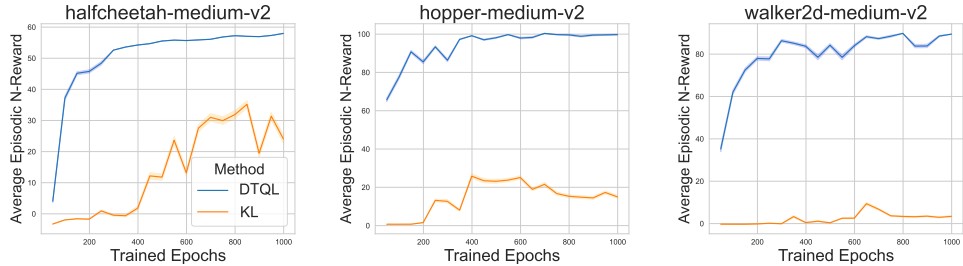

Figure 8: Training curves comparing policy learning with diffusion trust region loss and KL loss across three Gym medium tasks demonstrate that diffusion trust region regularization (DTQL) consistently outperforms KL-based behavior regularization in policy learning.

The results for Antmaze-v2 are shown in Table 6. Our observations indicate that, on average, our method achieves a higher score and exhibits significant performance improvements in complex Antmaze tasks, such as *antmaze-medium-diverse*, *antmaze-large-play*, and *antmaze-large-diverse*.

Table 6: The performance of Our methods and SOTA baselines on D4RL AntMaze-v2 tasks. Results for DTQL correspond to the mean and standard errors of normalized scores over 500 random rollouts.

| Antmaze | SRPO | Ours |
|---|---|---|
| antmaze-umaze-v2 | 97.1 | 92.6±1.24 |
| antmaze-umaze-diverse-v2 | 82.1 | 74.4±1.95 |
| antmaze-medium-play-v2 | 80.7 | 76±1.91 |
| antmaze-medium-diverse-v2 | 75.0 | **80.6**±1.77 |
| antmaze-large-play-v2 | 53.6 | **59.2**±2.19 |
| antmaze-large-diverse-v2 | 53.6 | **62**±2.17 |
| **Average** | 73.6 | **74.1** |

## F.4 Overall Training and Inference Time

In Table 7, we show the total training and inference wall time recorded on 8 RTX-A5000 GPU servers, which include all training epochs specified in Table 4 and the entire evaluation process. For evaluation, we test 10 trajectories for gym tasks and 100 trajectories for all other tasks.

Table 7: Total training and inference wall time for D4RL benchmarks

| Tasks | Overall Training and Inference Time | Training Epochs |
|---|---|---|
| halfcheetah-medium-v2 | 5.1h | 1000 |
| halfcheetah-medium-replay-v2 | 5.1h | 1000 |
| halfcheetah-medium-expert-v2 | 5.5h | 1000 |
| hopper-medium-v2 | 5.0h | 1000 |
| hopper-medium-replay-v2 | 5.4h | 1000 |
| hopper-medium-expert-v2 | 5.2h | 1000 |
| walker2d-medium-v2 | 4.9h | 1000 |
| walker2d-medium-replay-v2 | 4.9h | 1000 |
| walker2d-medium-expert-v2 | 4.9h | 1000 |
| antmaze-umaze-v0 | 3.3h | 500 |
| antmaze-umaze-diverse-v0 | 4.0h | 500 |
| antmaze-medium-play-v0 | 3.1h | 400 |
| antmaze-medium-diverse-v0 | 3.2h | 400 |
| antmaze-large-play-v0 | 2.3h | 350 |
| antmaze-large-diverse-v0 | 2.6h | 300 |
| antmaze-umaze-v2 | 3.3h | 500 |
| antmaze-umaze-diverse-v2 | 3.1h | 500 |
| antmaze-medium-play-v2 | 3.1h | 500 |
| antmaze-medium-diverse-v2 | 3.1h | 500 |
| antmaze-large-play-v2 | 3.3h | 500 |
| antmaze-large-diverse-v2 | 3.3h | 500 |
| pen-human-v1 | 1.4h | 300 |
| pen-cloned-v1 | 0.6h | 200 |
| kitchen-complete-v0 | 3.0h | 500 |
| kitchen-partial-v0 | 6.1h | 1000 |
| kitchen-mixed-v0 | 3.0h | 500 |

# G   Training Curves

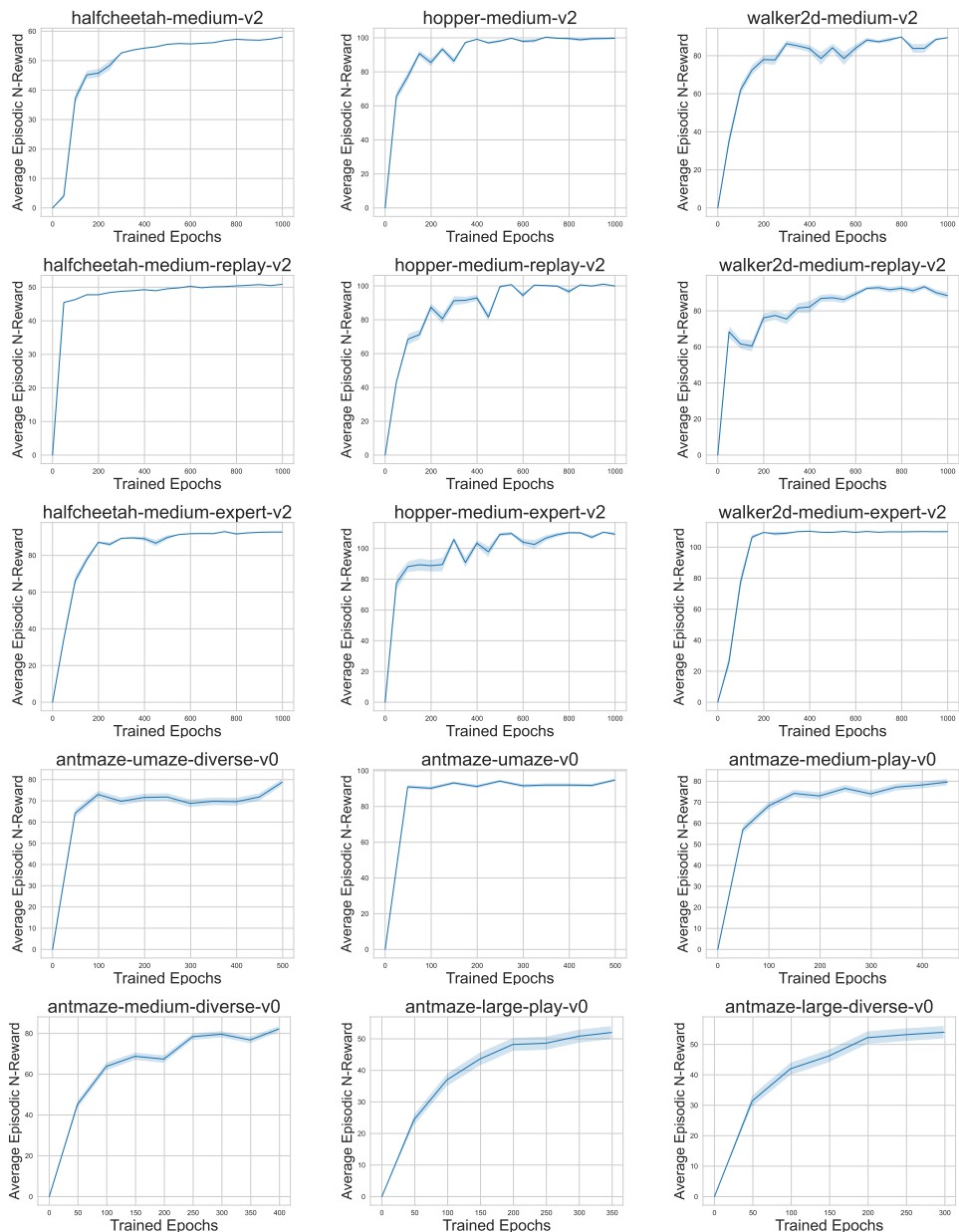

Figure 9: Training curves. Rewards evaluated after every 50 epochs.

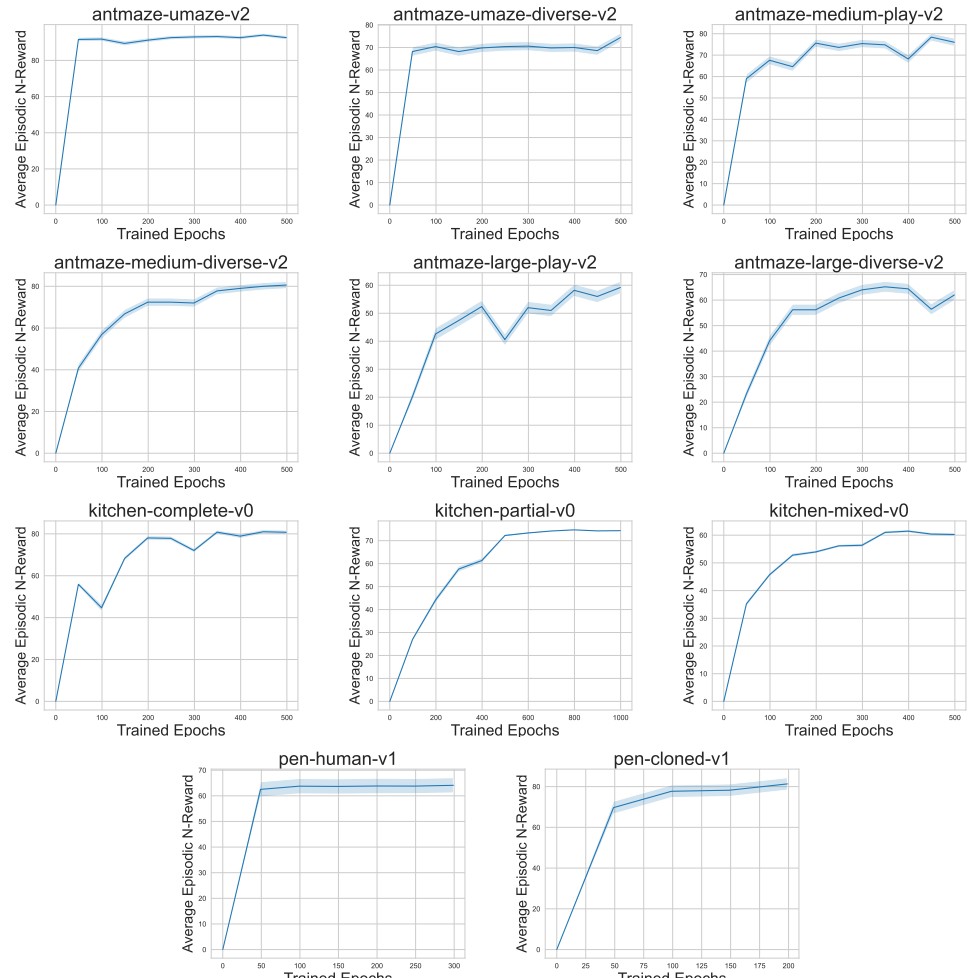

Figure 10: Training curves. Rewards evaluated after every 50 epochs.

