# OpenReview forum: "Diffusion Policies Creating a Trust Region for Offline Reinforcement Learning"
_NeurIPS.cc/2024/Conference — NeurIPS 2024 poster_

### Official Review · Reviewer_BozX · 2024-07-07

**Soundness:** 1
**Presentation:** 1
**Contribution:** 2
**Rating:** 4
**Confidence:** 4

**Summary:**

The paper proposes a new diffusion-based training loss for offline reinforcement learning, in which the empirical actions from the offline datasets are replaced with the actions sampled from a parameterized policy. The paper calls this new loss “trust region loss” and uses it as one of the objectives to minimize in the diffusion policy learning. The paper points out the mode seeking behavior induced by the minimization of such “trust region loss” and compares it with forward KL-regularization behavior cloning. The paper also evaluates the methods on D4RL datasets with some offline RL methods.

**Strengths:**

### One-step policy
Introducing a one-step policy seems a promising way to address the challenges in the diffusion sampling. This idea is nicely motivated and well-justified. The discussion on the mode-seeking behaviors is interesting, especially with the empirical results as an extra evidence to support.

**Weaknesses:**

I found the paper shares great similarities to the SRPO paper, in terms of the idea, the formulation, and the presentations. The empirical results of the proposed method are also close to SRPO method.
Beyond that, I have several other concerns:
### **Unknown trust region**
1. A bit surprisingly, **the paper gives no explicit definition of the crucial trust region used in Eq. (4)**, the diffusion-based trust region loss, even though trust region is the key for the whole paper. Particularly, the loss function defined by Eq. (4) comes directly from a simple modification of Eq. (3) by replacing the empirical action $a_0$ with the action generated by the policy $\pi_\theta$.  It is unclear what the trust region this simple modification induces.

2. Regarding “the loss effectively creates a trust region defined by the diffusion-based behavior cloning policy, within which the one-step policy $\pi_\theta$ can move freely”, the paper does not explain how “to generate actions (data) that reside in the high-density region of the data manifold specified by $\mu_\theta$ through optimizing $\theta$”. Note that $a_\theta$ generated by $\pi_\theta$ will be fed into $\mu_\phi$. **So a trivial solution of this optimization will be $\pi_\theta(\cdot|s) = a_0$, i.e., $\pi_\theta$ just returns $a_0$ for every $(a_0, s)\sim D$ as in Eq. (2). Then loss in Eq. (4) is exactly the same as Eq. (3)**. So how does the optimization of Eq. (4) avoid such trivial solutions?  Especially when the paper says Eq. (4) is to encourage mode seeking behaviors.

3. **the “one-step policy”, another important factor in the trust region, is introduced a bit inconsistently section 2.3 and 2.4**. I understand that Eq. (4) is to introduce a policy that seeks to model only one mode, i.e., one action only, from the training samples. But then line 133-137, “the generated action $a_\theta$ appear similar to in-sample actions and penalizes those that differ”, which means the action should be similar to any of the training samples to avoid penalties. Also, this “mode-seeking” contradicts with the statement “explore freely” in line 149: how does this policy move freely? And how the penalization is triggered? In addition, the paper mentioned an implicit parameterization of $\pi_\theta$: $a_\theta=\pi_\theta(s, \epsilon$. For this implicit parameterization, what’s the different between optimization Eq. (4) and the simple behavior cloning? One even more inconsistent description is given in line 183 “one-step Implicit policy”: why $\pi_\theta$ here is instantiated implicitly?


### A bit trivial discussion on mode seeking behaviors
**I found the discussion on mode seeking behaviors in Section 3 a bit trivial and pretty incremental as these have been pointed by SRPO already in its section 2.2**.

1. **the notations are not quite clear**: $p_{fake}$ and $p_{real}$ are not defined explicitly with respect to $\pi_\theta$ and $\mu_\phi$.

2. the forward KL divergence is mode-covering in that sense it learns a distribution that tries to cover all modes, while reverse KL divergence is mode-seeking, behaving similarly to the optimization of $L_{TR}$. This discussion has been provided by SRPO paper its equations (2) and (3). So what’s new? More importantly, **since the paper focuses more on the mode seeking behavior, why does it not compare against the reverse KL divergence minimization?**

3. **Theorem 1 tells very little about the mode seeking behavior of $\pi_\theta$ except the simple substitution of mode actions**. I don’t its importance in the training, especially the mode seeking behaviors.

### Hard to understand
**There are quite many sentences and descriptions with language issues or typos, which make the whole paper rather hard to understand**. Examples of language issues and typos:
* Line 192-193: “KL divergence is employed in this context, it is designed to recover all modalities of the data distribution”
* Line 199: “the loss ensures that the generated action .$a_\theta$ lies within the in-sample datasets”
* Line 288: “a Jacobian term” is different from what in Eq. (11).
* Line 126, “minimize $L_{PB}$ in Equation 4”. There is no such $L_{PB}$ term.
* Line 183, “is instantiates as an”

**Importantly, the hyperparameters used in the experiments are a bit hard to understand**. Particularly, in the experiments, the paper mentioned that “the variation between datasets significantly impacts the algorithm performance”. Then why did the paper still “employed the official SRPO code on Antemze-v0 and maintained identical hyperparameters used for Antmaze-v2”? Why wouldn’t the paper just report the performance on both versions, separating DQL, IDQL from SRPO?

**Questions:**

1. Can the authors explain which empirical results support that the assertion “diversity, while valued in image and 3D generation, is not essential in offline RL”? as speculated in Line 234 - 235.
2. Can the authors explain “discourage out-of-distribution sampling” in line 107?
3. What does it mean by “policy $\pi_\theta$ can move freely” in line 121?

---

> ### Author Rebuttal · Authors · 2024-08-05
>
> Thank you for your review. We believe there may have been some misunderstandings regarding our paper. We will strive to address your questions thoroughly and hope you will consider re-evaluating our work.
>
> ### Similarities
>
> Firstly, we have discussed some differences between our method and SRPO in the **Global Response**. Here we are addressing the reviewer's concerns about "presentation" and "empirical result" similarity.
>
> **The presentation.** We believe every excellent work, like DQL, SRPO, and IDQL, shares a similar logic in presenting the story in the offline RL setting. The presentation of our work humbly learns from these papers. Thus our presentation flow is not only similar to SRPO but also to DQL and IDQL, as these papers serve as excellent examples for us.
>
> **For empirical results.** Our method slightly outperforms SRPO on Gym tasks, achieving different SOTA results on various sub-tasks. For instance, SRPO scores higher on walker2d-m-e (114 vs. 110) but lower on hopper-m-e (100 vs. 109). Additionally, our method surpasses SRPO on antmaze-v0 tasks and excels on some antmaze-v2 sub-tasks like antmaze-l-p and antmaze-m-d (**Table 6**).
>
> ### Unknown trust region
>
> **1. Definition.** Formally, by Theorem 1, given any state $s$, we can define a trust region for action by the set {$a | \log p(a|s) \ge \text{threshold}$}, where $\log p(a|s)$ is approximated by the diffusion loss (Eq 4). The magnitude of the threshold can be tuned by the hyperparameter $\alpha$ during the optimization of the final loss (Eq. 5). Thank you for pointing this out; we will make this point clearer in the revision.
>
>  **2. Trivial solution.** For the reviewer's concerns about "move freely" and "how to generate actions...", we have tried to answer them in **Global Response**. For "trivial solution", such a trivial solution is only possible when dealing with discrete space control questions, and the offline datasets contain information for whole state spaces. However, for continuous control questions, such a trivial solution has a probability measure of 0 since both action and state spaces are continuous, and the size of the offline dataset is finite. Thus, this trivial solution does not apply to our setting. Moreover, even in the discrete case, the trivial solution you mentioned is essentially behavior cloning. When adding the Q function for guidance to improve the policy, $\pi$ will only return the action $a$ with the highest Q value for any given state $s$.
>
>  **3. Inconsistently.** We addressed the reviewer's concerns about "inconsistent" presentation in our global response section. Now, we aim to clarify "why $\pi_\theta$ is instantiated implicitly" in our approach. The reason for the implicit instantiation of $\pi_\theta$ in line 183 is due to the KL loss (Eq 8), which encourages mode seeking (Figure 2). The Gaussian policy has limited expressiveness; therefore, we make $\pi_\theta$ implicit to enhance its expressiveness and ability to seek more modes. It is important to note that Eq 8 is never used in our method; it is included only as a comparison baseline.
>
>
> ### Mode seeking behaviors
>
> **1. Trivial.** The discussion in Section 3 is primarily to demonstrate that our method indeed exhibits mode-seeking behaviors. While SRPO discusses mode-seeking for its own method, our focus here is to discuss the mode-seeking behavior specific to our approach. Therefore, we believe this discussion is necessary and not trivial.
>
> **2. Notations.** Constrained by the length of the paper, we do not demonstrate all details about the algorithm in Section 3 since it is not our main contribution. However, we did mention in line 191 that more details are deferred to Appendix D. For full details, please refer to Variational Score Distillation [33], Diff-Instruct [23], and Distribution Matching Distillation [37]. These reference indices are consistent with those used in our paper.
>
> **3. Reverse KL divergence minimization.**  This was addressed by SRPO, and since our paper thoroughly compares with SRPO, we believe we have sufficiently addressed the comparison with reverse KL divergence.
>
> **4. Theorem 1.** Please refer to our global response.
>
> ### Hard to understand
>
> **1. Typos.** Thank you for the careful review and valuable suggestions. We will incorporate these corrections in the revised manuscript.
>
> **2. Experiments.** We have contacted the authors of SRPO by email, and they acknowledged that there is a version mismatch between SRPO and other baselines in Antmaze v0 and v2. Since SRPO does not provide hyperparameters for Antmaze v0, we used the same hyperparameters as for v2 and reported them in the main table. Additionally, as illustrated in line 262 and in **Table 6**, we also compare our method with SRPO on **Antmaze v2**, where we use same hyperparameters as those in Antmaze v0 for our method. The SRPO results are taken from the original paper for a fair comparison. And separating table is a good idea. We can separate SRPO from the main table and create an additional table to compare results in the revision, which will only include Antmaze-v2 results if the reviewer believes it is better to present in this way.
>
> ### Responses to Questions
>
> For Questions 2 and 3, we believe they have been covered in the global response. For Question 1, in offline RL, typically after training and during interaction with the environment, for any given state, we want the agent to generate the action with the highest Q value. Therefore, diversity, in the sense of generating a batch of diverse actions for a given state, is not essential. As long as the action has a high Q value, the diversity of the generated actions does not contribute to increasing the final cumulative reward. A good example is the deterministic policy, which, for a given state, generates only one action. Although it lacks diversity, it can still achieve a high Q value.

---

> > ### Comment · Reviewer_BozX · 2024-08-11
> > **Response to rebuttal**
> >
> > Thanks for providing the rebuttal response. I still have concerns and some of my questions are also not answered directly. I do think the paper in its current version needs some substantial improvement. I would thus keep my original assessment.

---

> > > ### Author Response · Authors · 2024-08-11
> > >
> > > Thank you for your reply. We have made every effort to address your questions and believe our rebuttal has thoroughly covered all of your concerns. We have elaborated on the differences in motivation, provided detailed explanations of the theorem proof, clarified the theorem explanation, and presented comprehensive empirical results. Each of your questions has been answered point by point. We believe we have effectively addressed your concerns, as we did with Reviewer R2od and Reviewer Rh1M, both of whom were satisfied with our responses and voted for acceptance.
> > >
> > > However, since you still have some concerns and mentioned that some questions were not directly addressed, could you please clarify which points remain unanswered or which aspects are still of concern? We would be happy to engage in further discussion on these matters.

---

### Official Review · Reviewer_Rh1M · 2024-07-10

**Soundness:** 3
**Presentation:** 2
**Contribution:** 3
**Rating:** 7
**Confidence:** 4

**Summary:**

This paper introduces Diffusion Trusted Q-Learning (DTQL) for offline reinforcement learning. DTQL employs a dual-policy representation, a diffusion policy trained by behaviour cloning, and a one-step Gaussian policy trained by RL and distilling the diffusion model. Specifically, DTQL introduces a diffusion trust region loss for the distillation. Such a scheme allows both one-step generation, and achieves strong performance on the D4RL benchmark.

**Strengths:**

1/ The proposed method is interesting.

2/ The proposed DTQL is well-motivated. It addresses one of the big issues of diffusion policy, i.e., the multi-step inference, through policy distillation.

3/ The paper has conducted solid experiments, for both intuitive understanding and empirical results on benchmarks.

**Weaknesses:**

Overall I don’t see many weak points of this work.

1/ The distilled policy captures some certain modes, but is actually not multi-modal. It also missed certain modes as in Fig. 2, which is inconsistent with what the authors have claimed in the caption. I would suggest the authors to be careful and avoid over-claiming.

2/ The writing has certain room for improvement. The introduction could be reorganised for better clarity. There are some typos, e.g., in Line 183, “instantiates”, and inconsistent wording, e.g., “cooperative policy” and “dual policy”.

**Questions:**

1/ It seems to me from Fig. 2 that this approach encourages interpolation of the distribution. Have you tried to design tasks to validate such behaviours?

2/ It is a bit unclear how to guarantee the distilled Gaussian policy captures all relevant information of the diffusion process, as its policy class naturally limits the expressiveness. Could you provide more explanations for this?

**Limitations:**

The experiments are done only on simple D4RL benchmarks. The policy distribution is quite simple and tasks are relatively short horizon. As a result, I’m not fully convinced when applied to more challenging scenarios, distilling the diffusion policy into a Gaussian with trust-region loss can still work given more complex action distributions.

---

> ### Author Rebuttal · Authors · 2024-08-06
>
> We appreciate the reviewer's recognition of our method's novelty, motivation, and empirical performance. Thank you for the careful review.
>
> ### Responses to Weakness 1.
>
> Thank you for the good suggestion. You are absolutely right that some modes are also missed by $\mathcal{L}_{\text{KL}}$. We will modify the caption accordingly in the revision to accurately reflect this.
>
> ### Responses to Weakness 2.
>
> Thank you for pointing this out and for your valuable suggestions. We will continue polishing our paper to enhance clarity. We apologize for the oversight in Line 183 with the word "instantiates." We will correct this and thoroughly review the entire paper to eliminate any other typographical errors. Additionally, we will correct the inconsistent wording and change "cooperative policy" to "dual policy" to provide clearer communication to the readers.
>
>
> ### Responses to Question 1.
>
> Thank you for the insightful observation. We hypothesize that this behavior is due to neural networks' tendency to learn continuous functions, which can lead to the connection of some modes with artificial dots. This phenomenon also occurs when using diffusion models to mimic distributions with isolated density modes. For example, in Figure 2 of DQL [1], there are similar artifacts in the interpolation of modes when we use pure diffusion policy to do behaviour cloning.
>
> Yes, we have also designed an RL algorithm based on $\mathcal{L}_{\text{KL}}$, as shown in Figure 3 and Appendix G.2. We suspect that interpolation may cause some out-of-distribution action generation, which could explain the performance drop observed in the learning curve in Figure 8.
>
> [1] Wang, Z., Hunt, J. J., \& Zhou, M. (2022). Diffusion policies as an expressive policy class for offline reinforcement learning. arXiv preprint arXiv:2208.06193.
>
> ### Responses to Question 2.
>
> Thank you for the question. We are happy to make some clarification on this point.
>
> 1. We don't expect the Gaussian policy to capture all the information from the diffusion model due to the unimodal nature of Gaussian policies. However, the Gaussian policy is sufficient to capture the optimal mode, which has the highest Q value and is usually unimodal. For example, in Figure 2 of DQL [1], even though the diffusion policy initially captures all behavior modes, it eventually focuses only on one optimal mode after policy improvement (guided by Q value). Thus, we believe that in an offline RL setting, the Gaussian policy is able to capture one optimal behavior mode when constrained by the diffusion policy.
>
> 2. We chose the Gaussian policy because it is widely used in RL settings, such as in IQL [2] and SAC [3], making it a good starting point to demonstrate the performance of our algorithm. However, our algorithm can accommodate any one-step policy. One possible extension is to use an implicit policy parameterized by a neural network to enhance the expressiveness of the policy and cover multimodal distributions if necessary.
>
> [2] Kostrikov, I., Nair, A., \& Levine, S. (2021). Offline reinforcement learning with implicit q-learning. arXiv preprint arXiv:2110.06169.
>
> [3] Haarnoja, T., Zhou, A., Abbeel, P., \& Levine, S. (2018, July). Soft actor-critic: Off-policy maximum entropy deep reinforcement learning with a stochastic actor. In International conference on machine learning (pp. 1861-1870). PMLR.
>
> ### Responses to Limitations.
>
> Thanks for the discussion. We evaluate our algorithm using widely used benchmarks for fair comparison with other offline RL algorithms. We acknowledge that the Gaussian policy may not have sufficient expressiveness for complex tasks. However, our algorithm framework can accommodate any one-step policy. Therefore, any policy, such as an implicit policy defined by passing random noise through a deep neural network, can potentially be used in our algorithm to handle more complex tasks. This represents an interesting future direction to explore how these diffusion-based RL algorithms perform in more complex settings.

---

> > ### Comment · Reviewer_Rh1M · 2024-08-09
> >
> > Thank you for the detailed reponse. My concerns are addressed. I would keep my original rating and vote for acceptance.

---

> > > ### Author Response · Authors · 2024-08-10
> > >
> > > We sincerely thank the reviewer for their constructive suggestions and positive feedback. We will implement the modifications as discussed, and we appreciate the reviewer’s input, which will help us further enhance our paper.

---

### Official Review · Reviewer_Px6z · 2024-07-12

**Soundness:** 2
**Presentation:** 2
**Contribution:** 2
**Rating:** 4
**Confidence:** 3

**Summary:**

The paper introduces Diffusion Trusted Q-Learning (DTQL), a novel approach for offline reinforcement learning that enhances performance and efficiency. DTQL utilizes a dual policy framework to bridge the gap between diffusion policies and one-step policies, improving expressiveness without the need for iterative denoising sampling. By incorporating a diffusion trust region loss, DTQL ensures stability and effectiveness in training. The method outperforms traditional distillation methods in various scenarios, offering a promising solution for offline RL challenges.

**Strengths:**

* To my knowledge this paper presents a new approach to make Diffusion policy computationally efficient compared with previous method.

* Based on the experimental results, it is evident that DTQL outperforms the previous method in terms of performance.

**Weaknesses:**

* This is an incremental work that relies on Diffusion Q-learning (DQL) and adds a weighted loss to the diffusion model objectives(Equation 4) to accelerate model inference, without clearly explaining the necessity of this combination.

* The explanation of the proposed method in Section 2 is too brief, with details only provided in subsection 2.3. Additionally, subsection 2.3 does not clearly explain the meaning of Trust Region in Equation 4. The subsequent Theorem and Remark do not effectively justify the necessity of using this update formula. The overall logic lacks coherence.

* The introduction of other methods for accelerating training and inference in diffusion-based policy learning in Section 3 is overly lengthy and lacks relevance to the methods used in this paper. It is recommended to move most of this content to the appendix.

**Questions:**

* How should the term "trust region" in the article's title and method name be understood?

* How is the weight function $w(t)$ in Equation 4 set in this paper, and why is it configured in this manner? A detailed discussion is needed.

* How does DTQL eliminate the need for iterative denoising sampling during both training and inference, making it computationally efficient?

**Limitations:**

* The writing of this paper has issues—there is a lack of coherence between the motivation, method, and experiments. Section 2 provides insufficient detail on the proposed method and does not effectively explain why it improves computational efficiency. Additionally, Section 3 dedicates too much space to the comparison experiments replacing $\mathcal{L}_{\mathcal{TR}}$, lacking a comparative analysis with the baseline method DQL.

* The method presented in this paper is merely a simple combination of previous works [1] and [2], which is lack of novelty.

[1] Wang, Zhendong, Jonathan J. Hunt, and Mingyuan Zhou. "Diffusion Policies as an Expressive Policy Class for Offline Reinforcement Learning." The Eleventh International Conference on Learning Representations.

[2 ] Kingma, Diederik, and Ruiqi Gao. "Understanding diffusion objectives as the elbo with simple data augmentation." Advances in Neural Information Processing Systems 36 (2024).

---

> ### Author Rebuttal · Authors · 2024-08-06
>
> Thank you for your review. We appreciate the time and effort you have dedicated to evaluating our work. It seems there may have been some misunderstandings regarding the main points of our paper. We will do our best to clarify these issues and hope you will consider re-evaluating our work.
>
> ### Responses to Weakness 1,2,3
>
> We try to address the reviewer's concerns in the **Global Response**, where we introduce the main goal, contribution, and presentation logic of our paper. We believe this global response adequately addresses the reviewer's concerns regarding the weaknesses. If the reviewer still has any questions about these aspects, we are happy to discuss them further.
>
> ### Response to Question 1
>
> **1.From Toy Example:** Let us illustrate this with toy examples. In Figure 2, the first column shows the offline action data distribution, and the second column shows the trust region loss for different actions. We can observe that actions lying in the high-density region of the raw data manifold have less trust region loss. This means that actions similar to in-sample data are trusted and have lower loss. In contrast, actions outside the in-sample data manifold, as shown in the second and third columns, have large trust region loss. This visualization demonstrates that for a given fixed diffusion model $\mu_\phi$, the loss $E[||\mu_\phi(a_\theta+\sigma_t\epsilon,t|s)-a_\theta||^2]$ can be used to check whether the action $a_\theta$ generated by the one-step policy $\pi_\theta$ is similar to or different from in-sample actions. If the generated data is similar to in-sample data, it is trusted and has a lower trust region loss.
>
> **2.From Theoretical Perspective:** From Theorem 1, the trust region can be defined by the conditional log likelihood. Specifically, for any given state $s$, a trust region of action is a set {$a|\log p(a|s)\geq\text{threshold}$} where the conditional log likelihood is approximated by conditional diffusion loss (Eq 4). The magnitude of the threshold can be tuned by the hyperparameter $\alpha$ during the optimization of the final loss (Eq. 5).
>
> **3.Further Discussion:** The discussion about the "trust region" is also covered from Line 117 to Line 122. If the reviewer has any further questions about understanding this loss, we are happy to discuss it in more detail.
>
>
>
> ### Response to Question 2
>
> As our goal is not to design a new weight schedule for the diffusion model, we are using the state-of-the-art diffusion weight schedule, EDM [3], which is discussed in Section 4.3 and Appendix C for completeness. However, other diffusion training schedules, such as VP and VE [4], can also be accommodated within our algorithm framework.
>
> [3] Karras, T., Aittala, M., Aila, T., \& Laine, S. (2022). Elucidating the design space of diffusion-based generative models. Advances in neural information processing systems, 35, 26565-26577.
>
> [4] Song, Y., Sohl-Dickstein, J., Kingma, D. P., Kumar, A., Ermon, S., \& Poole, B. (2020). Score-based generative modeling through stochastic differential equations. arXiv preprint arXiv:2011.13456.
>
> ### Response to Question 3
>
> In brief, we do not use a diffusion policy to generate actions, thereby avoiding iterative denoising. A comprehensive explanation of this point is discussed in the **global response**. If the reviewer has any further questions about this point, we are happy to discuss them.
>
> ### Response to Limitation 1
>
> We are confident that by considering the **global response** and gaining a more comprehensive understanding of our paper, the logic and presentation will become clearer and more accessible for readers. We kindly request that the reviewer re-evaluate our paper in light of these clarifications.
>
> ### Response to Limitation 2
>
> As discussed in the **global response**, our paper is not the same as DQL [1]. On the contrary, we introduce a completely different training scheme to address the drawbacks in DQL. Additionally, [2] is used to support Theorem 1, and we believe that our use of this reference does not diminish the novelty of our new offline RL method design in any way.

---

> > ### Comment · Reviewer_Px6z · 2024-08-13
> > **Response**
> >
> > I thank the authors for the detailed rebuttal. Most of by concerns has been addressed. I would like to raise the score. However, the organization of this paper can be improved. For the current form, the reviewer found it not easy to follow.

---

> > > ### Author Response · Authors · 2024-08-13
> > >
> > > Thank you very much for your thoughtful response. We are pleased to hear that most of your concerns have been well addressed. We sincerely appreciate your suggestion regarding the writing, and we will carefully consider making adjustments to the structure of the presentation to further promote understanding. Should you have any other recommendations in presentation, please let us know and we are happy to involve that.
> > >
> > > If there are no further technical questions and given our significant improvement in both efficiency and performance, we would respectfully ask if you might consider re-evaluating our paper with a view towards a more positive recommendation for acceptance.

---

### Official Review · Reviewer_R2od · 2024-07-12

**Soundness:** 3
**Presentation:** 3
**Contribution:** 3
**Rating:** 6
**Confidence:** 3

**Summary:**

This paper introduces DTQL, a novel offline reinforcement learning (offline RL) method. With a newly introduced diffusion trust region loss, DTQL constrained policy update within a predefined trust region near a diffusion policy trained by behavior cloning (BC). Through the empirical experiments, DTQL demonstrated improved performance against BC and offline RL baselines, including recent baselines that utilize diffusion models in offline RL settings.

**Strengths:**

1. The diffusion-based trust region loss looks novel and interesting.

2. The visualization of trust region loss and the toy-tasks provide an intuitive explanation of the proposed algorithm.

3. The empirical experiments show improved performance compared to standard offline RL methods and faster inference time than most offline RL with diffusions. The reviewer appreciates that the author also shows the results where DTQL performs worse than IDQL (among the Antmaze tasks), which provides a more comprehensive view of the algorithm's performance.

4. The paper is easy to follow, and the connections and differences to related works are clearly outlined.

**Weaknesses:**

1. The performance improvements compared with DQL, IDQL, and SRPO are not very significant; in some experiments (Antmaze), IDQL outperformed the proposed algorithms.

2. The ablation studies only show one seed result, which reduces the results' plausibility, especially for Figure 5 (a), where the learning curves are quite noisy.

**Questions:**

1. The reverse KL objective should also be mode-seeking instead of mode-averaging. I do not quite understand why the KL regularization does not show this property in the example provided in Figure 3. As in the SRPO[1], a similar illustration was shown in Figure 5. Could the author provide more insights on what caused this difference in behavior?

2. When DTQL only utilizes four layers of MLP and SRPO requires several resnet blocks, why is SRPO inference faster than DTQL?

[1] Chen H, Lu C, Wang Z, Su H, Zhu J. Score regularized policy optimization through diffusion behavior. arXiv preprint arXiv:2310.07297. 2023 Oct 11.

**Limitations:**

The limitations were discussed in the conclusion sections, along with possible future works.

---

> ### Author Rebuttal · Authors · 2024-08-05
>
> We appreciate the reviewer's recognition of novelty, presentation and performance improvement of our paper. Below, we provide detailed clarifications and answers to solve the remaining concerns.
>
> ### Responses to Weakness 1.
>
> We first want to emphasize that one major contribution of DTQL is for speeding up Diffusion-QL. We proposed a dual policy training approach and achieved dramatically improvement in terms of both training and inference efficiency.
>
> For D4RL benchmark performance, we provide more clarifications below.
> 1. Compared with DQL, our major baseline, our method demonstrates dramtically speeding up, superior average score and greater stability. We expand the explanation for stability here. As shown in Figure 8 of [2], the training curve of DQL is less stable due to the challenges in computing gradients of the Q-value function while backpropagating through all diffusion timesteps, which may result in a vanishing gradient problem. In contrast, we propose a dual policy trainig approach, and train a one-step Gaussian policy with trust region defined by a diffusion policy. This avoids the gradient vanishing issue and mitigates training instability, as shown in the learning curve in Appendix H.
> 2. DTQL outperforms IDQL in Gym and slightly underperforms it in Antmaze. However, IDQL is **10x** slower than us in inference.
> 3. Regarding SRPO, our method not only scores higher on Gym tasks but also surpasses SRPO on Antmaze-v0 as shown in Table 1 (from 30.1 to 73.6) and Antmaze-v2 as shown in Appendix G.3.
>
> [2] Lu, C., Chen, H., Chen, J., Su, H., Li, C., \& Zhu, J. (2023, July). Contrastive energy prediction for exact energy-guided diffusion sampling in offline reinforcement learning. In International Conference on Machine Learning (pp. 22825-22855). PMLR.
>
> ### Responses to Weakness 2.
>
> Thank you for your suggestion. We have updated the ablation study to include results across 5 random seeds in the PDF of global response, along with the standard error. We would like to make the following clarifications:
> 1. The blue line in Figure 5 (a), only used for ablation study, is quite noisy because it does not include the entropy term. After adding the entropy term, the orange line for the reward curve of the proposed algorithm becomes much more stable.
> 2. We acknowledge that the orange line in the antmaze-large-diverse-v0 appears noisy. However, this level of variations is common in the offline RL setting and is, in fact, found to be more stable than DQL.
>
> ### Responses to Question 1.
>
> Thanks for giving us opportunity to explain about this, and we will also make more discussion about it in the revision. In sum, KL regularization in Figure 3 is not the same as reverse KL used in SRPO. In Figure 3, we are using the loss from Diff-Instruct [3], Variational Score Distillation [4]  and Distribution Score Matching [5]. The detailed algorithm is described in Appendix D. An analysis of the gradients shows distinct differences between the Score Distillation Sampling (SDS) loss in SRPO [1] and our loss:
>
> - As discussed beginning at line 220, for SRPO [1], we have:
> $$
> \nabla_\theta L_{\text{SDS}} =E_{t,s,\epsilon}\left[w(t)(\epsilon_\phi(z_t,t|s) - \epsilon)\frac{\partial z_t}{\partial \theta}\right]
> $$
>  - In contrast, the loss we used in Figure 3 is:
> $$
> \nabla_\theta L_{\text{KL}} = E_{t,s,\epsilon}\left[w(t)(\epsilon_\phi(z_t,t|s) - \epsilon_{\text{fake}}(z_t,t|s))\frac{\partial z_t}{\partial \theta}\right]
> $$
> where $\epsilon_{\text{fake}}$ is the score function we learned from the current policy. Thus, the loss we used in Figure 3 provides an updating direction that can be rather “fine” and “sharp” due to the difference between pretrained $\epsilon_\phi$ and $\epsilon_{\text{fake}}$. This approach is encouraging for covering more modes (as shown in Figure 2) and differs from the reverse KL loss (SDS loss) in SRPO. A more detailed comparison has been discussed in [4].
>
> [3] Luo, W., Hu, T., Zhang, S., Sun, J., Li, Z., \& Zhang, Z. (2024). Diff-instruct: A universal approach for transferring knowledge from pre-trained diffusion models. Advances in Neural Information Processing Systems, 36.
>
> [4] Wang, Z., Lu, C., Wang, Y., Bao, F., Li, C., Su, H., \& Zhu, J. (2024). Prolificdreamer: High-fidelity and diverse text-to-3d generation with variational score distillation. Advances in Neural Information Processing Systems, 36.
>
> [5] Yin, T., Gharbi, M., Zhang, R., Shechtman, E., Durand, F., Freeman, W. T., \& Park, T. (2024). One-step diffusion with distribution matching distillation. In Proceedings of the IEEE/CVF Conference on Computer Vision and Pattern Recognition (pp. 6613-6623).
>
>
>
> ### Responses to Question 2.
>
> Thank you for your observation. We will clarify this point in the revision.
> 1. SRPO uses a deterministic policy, whereas our method employs a stochastic policy. Consequently, after the network forward pass, we need to resample to generate an action, which adds to the inference time.
> 2. Additionally, we utilize the (stochastic) max Q trick in inference, as in DQL, which involves generating $N$ candidate actions for a given state and then selecting an action randomly with a weight proportional to $\exp{Q(s,a)}$.
>
> Therefore, these two factors—stochastic resampling and max Q —contribute to making our inference slightly slower than the deterministic policy used in SRPO. We will provide a more detailed explanation of this point in the revised version.

---

> > ### Comment · Reviewer_R2od · 2024-08-09
> >
> > I appreciate the authors' detailed response and extra experiments, and I have no more questions at this stage.
> > I suggest the authors include the SRPO style reverse KL in Figure 3 in the final version of this work to provide a more intuitive visual understanding of the differences between them.
> > I will keep my score and lean towards a positive outcome of this work.

---

> > > ### Author Response · Authors · 2024-08-10
> > >
> > > We appreciate the reviewer’s constructive and valuable suggestions. We will incorporate an SRPO-style example into Figure 3 as recommended in the revised manuscript.

---

### Author Rebuttal · Authors · 2024-08-06

We thank the reviewers for their valuable feedback. We believe some aspects of our paper may have been misunderstood, particularly by **Reviewer Px6z regarding the goal, contribution, and logic of our paper**, and by **Reviewer BozX who questioned the differences with SRPO**. Below, we focus on addressing these two concerns.

## Goal, Contribution, and Logic

The goal of our paper is to accelerate training and inference for diffusion-policy-based offline RL methods, such as DQL, while also increasing training stability. Previous baselines, like DQL, generate actions through iterative denoising during both training and inference phases. This process cannot be computed in parallel, making training and inference time-consuming. Additionally, since actions are generated by iterative denoising, gradient computation requires backpropagation through all diffusion time steps. This can result in a vanishing gradient problem, undermining training stability.

Our paper aims to address these issues with two main objectives:

1. Leverage diffusion policy and enhance performance.

2. Avoid iterative denoising for action generation, thus accelerating training and inference and enhancing stability.

To this end, we propose a dual policy method, containing one diffusion policy and one one-step Gaussian policy. The diffusion policy is not used to generate actions directly, while the Gaussian policy generates actions for both training and inference. **This one-step policy eliminates the need for iterative denoising, thus speeding up training and inference.** Additionally, since actions are not generated through iterative denoising, the vanishing gradient issue is mitigated, resulting in more stable training.

We introduce a Trust Region loss (Eq. 4) to bridge the two policies. After performing pure behavior cloning, using actions from the dataset to train diffusion policy $\mu_\phi$, we generate actions from the one-step policy. We then reuse the diffusion loss calculated by $\mu_\phi$ to evaluate how much the generated actions (from the one-step policy) deviate from the existing dataset. Figure 1 demonstrates that if the generated action is similar to in-sample actions, the Trust Region loss is small; if the generated action deviates significantly from in-sample actions, the Trust Region loss is large. Thus, the Trust Region loss uses pretrained $\mu_\phi$ to measure whether the actions generated by the one-step policy $\pi$ are similar to or different from the in-sample data.

Then adding Q function, we derive the final loss in Eq. 5. Since the one-step policy $\pi$ is the actual policy interacting with the environment, and we only use $\mu_\phi$ to calculate the diffusion loss without reverse sampling, training and inference are accelerated. The first component in Eq. 5 regularizes the generated action to not deviate far from the dataset, and the second component maximizes the Q value.

## Difference with SRPO

Some concerns about our method is similar to SRPO. While we aim to solve the same problem of diffusion policy as introduced above, our approach differs significantly from SRPO in terms of idea, loss formulation, loss explanation, theorem, and empirical results, as elaborated below:

### 1. Idea and loss formulation difference

We believe the intuition behind our loss $L_{\text{TR}}$ (Eq. 4) fundamentally differs from the reverse KL-based SRPO loss. The starting point and motivation of our paper is to reuse the diffusion loss as a regularization, which helps us determine whether the generated action is far from the in-sample dataset. Here, the diffusion policy acts more like a **detector** to detect out-of-distribution actions. In contrast, the motivation of SRPO is to use reverse KL to distill information from diffusion policies, where the diffusion policy serves more as a **distillation target** rather than a detector. The loss formulations are also evidently different. Let $z_t = \alpha_t a_\theta + \sigma_t \epsilon$ where $a_\theta \sim \pi_\theta$. The formulations of the losses for SRPO and our method are clearly different.:

$$
L_{\text{SRPO}} = E_{s,\epsilon,a_\theta\sim\pi_\theta}[\log\pi_\theta(a_\theta|s) - \log \mu_\phi(a_\theta|s)]
$$

$$
L_{\text{TR}} = E_{s,\epsilon,a_\theta\sim\pi_\theta}[||\mu_\phi(z_t|s) - a_\theta||_2^2]
$$


### 2. Theorem explanation difference

In addition to Theorem 1, which demonstrates that our loss encourages the generated action $a_\theta$ to lie in the high-density region of the offline data, we would like to further clarify the differences:

- During the training of the diffusion model, we aim to optimize $\phi$ to minimize the negative log likelihood $E_{a\sim D}[-\log\mu_\phi(a|s)]$. However, we use a variational upper bound, defined as the diffusion training loss in Eq. 2, for tractable optimization (Theorem 1 and [1]). During the optimization of policy $\pi_\theta$, we adapt this diffusion training loss from Eq. 4 as our trusted region loss $L_{\text{TR}}$, which acts as a variational upper bound for the negative log likelihood $E_{a\sim \pi_{\theta}}[-\log\mu_\phi(a_{\theta}|s)]$. Therefore, there is no inconsistency between the diffusion training and diffusion-based regularization in our method.

- We further observe that SRPO incorporates $E_{a\sim \pi_{\theta}}[-\log\mu_\phi(a_{\theta}|s)]$ as the second term in its KL-based theoretical loss. The other component of the SRPO KL loss is $E_{a\sim \pi_{\theta}}[\log\pi_\theta(a_\theta|s)]$, which is notably absent in our loss formulation. The theoretical loss of SRPO is intractable to optimize, necessitating an approximation that substitutes $a_\theta$ with $z_t$. This substitution further differentiates the SRPO loss from our DTQL loss.

[1] Song et al. "Maximum likelihood training of score-based diffusion models." NeurIPS 2021.

### 3. Empirical performance difference

Due to character limitations, the discussion will be provided in our response to the specific reviewer.

---

### Decision · Program_Chairs · 2024-09-25

**Decision:**

Accept (poster)

**Comment:**

Most of the criticism regarding this submission centered on the clarity of its positioning and the interpretation of the trust region. As the Area Chair, I believe the authors have addressed these concerns satisfactorily in their rebuttal, and I therefore recommend acceptance. However, I strongly encourage the authors to undertake a significant revision of the paper to ensure it fully realizes its potential and provides maximum value to the community.